# The transcription factors Runx3 and ThPOK cross-regulate acquisition of cytotoxic function by human Th1 lymphocytes

Yasmina Serroukh[1†], Chunyan Gu-Trantien[1†], Baharak Hooshiar Kashani[1†], Matthieu Defrance[2], Thien-Phong Vu Manh[3], Abdulkader Azouz[1], Aurélie Detavernier[1], Alice Hoyois[1], Jishnu Das[4,5], Martin Bizet[2], Emeline Pollet[3], Tressy Tabbuso[1], Emilie Calonne[2], Klaas van Gisbergen[6], Marc Dalod[3], François Fuks[2], Stanislas Goriely[1], Arnaud Marchant[1]*

[1]Institute for Medical Immunology, Université Libre de Bruxelles, Charleroi, Belgium; [2]Laboratoire d'Epigénétique du Cancer, Université Libre de Bruxelles, Bruxelles, Belgium; [3]Centre d'Immunologie de Marseille-Luminy 13288, Aix Marseille Université UM2, Marseille, France; [4]Ragon Institute of MGH, MIT and Harvard University, Cambridge, United States; [5]Department of Biological Engineering, Massachusetts Institute of Technology, Cambridge, United States; [6]Department of Haematopoiesis, Sanquin Research and Landsteiner Laboratory, Amsterdam, Netherlands

*For correspondence:
arnaud.marchant@ulb.ac.be

†These authors contributed equally to this work

Competing interests: The authors declare that no competing interests exist.

**Abstract** Cytotoxic CD4 ($CD4_{CTX}$) T cells are emerging as an important component of antiviral and antitumor immunity, but the molecular basis of their development remains poorly understood. In the context of human cytomegalovirus infection, a significant proportion of CD4 T cells displays cytotoxic functions. We observed that the transcriptional program of these cells was enriched in CD8 T cell lineage genes despite the absence of ThPOK downregulation. We further show that establishment of $CD4_{CTX}$-specific transcriptional and epigenetic programs occurred in a stepwise fashion along the Th1-differentiation pathway. In vitro, prolonged activation of naive CD4 T cells in presence of Th1 polarizing cytokines led to the acquisition of perforin-dependent cytotoxic activity. This process was dependent on the Th1 transcription factor Runx3 and was limited by the sustained expression of ThPOK. This work elucidates the molecular program of human $CD4_{CTX}$ T cells and identifies potential targets for immunotherapy against viral infections and cancer.
DOI: https://doi.org/10.7554/eLife.30496.001

## Introduction

The thymic differentiation of helper CD4 and cytotoxic CD8 T lymphocytes results from the opposite activity of key transcription factors (TF) including ThPOK and Runx3 repressing the expression of CD8 and CD4 T cell lineage genes, respectively. After emigration from the thymus, naive CD8 and CD4 T cells maintain the expression of Runx3 or ThPOK, respectively, suggesting that the lineage-defining role of these TF is also active in the periphery (*Vacchio and Bosselut, 2016*). However, the repression of a cytotoxic program in peripheral CD4 T cells is not absolute as these cells can acquire perforin-dependent cytotoxic activity (*Appay et al., 2002a*; *van de Berg et al., 2008*; *Cheroutre and Husain, 2013*).

Initially considered as a phenomenon of peripheral importance, the acquisition of cytotoxic function by CD4 T cells is now recognized as a key component of immunity against viruses and tumors (*Swain et al., 2012*) and correlates with positive outcome in multiple human and animal models (*Brown et al., 2012*; *Wilkinson et al., 2012*; *Johnson et al., 2015*; *Ma et al., 2015*; *Weiskopf et al., 2015*; *Verma et al., 2016*; *Xie et al., 2010*; *Quezada et al., 2010*; *Fu et al., 2013*). Beside their role in viral infections and cancer, cytotoxic CD4 ($CD4_{CTX}$) T cells may also have a pathogenic role in chronic inflammatory disorders (*Dumitriu, 2015*).

The development of $CD4_{CTX}$ T cells remains incompletely understood. In humans, $CD4_{CTX}$ T cell function is a hallmark of terminally differentiated antigen-experienced cells producing large amounts of gamma interferon (IFNγ) and low levels of interleukin-2 (IL-2) (*Appay et al., 2002a*; *Casazza et al., 2006*). This suggests that $CD4_{CTX}$ T cell differentiation might be induced by the prolonged stimulation of Th1 lymphocytes. On the other hand, recent studies suggest that $CD4_{CTX}$ T cells form a distinct lineage emerging from precursors expressing class-I restricted T cell-associated molecule (CRTAM) (*Takeuchi et al., 2016*). Several TF probably contribute to the differentiation of $CD4_{CTX}$ T cells within or outside the Th1 pathway. In the mouse intestine, the acquisition of cytotoxicity by CD4 T cells is associated with the down regulation of ThPOK and the upregulation of Runx3 (*Mucida et al., 2013*; *Reis et al., 2013*; *Sujino et al., 2016*). In CD8 T cells, the cytotoxic program is activated by the cooperation of TF from the T-box family (T-bet and Eomes) and Runx3, (*Cruz-Guilloty et al., 2009*; *Pearce et al., 2003*). In mouse models of cancer or neuroinflammation, Eomes was required for the induction of $CD4_{CTX}$ T cells (*Curran et al., 2013*; *Raveney et al., 2015*). In murine influenza infection, Blimp1 was required for $CD4_{CTX}$ T cell differentiation (*Marshall et al., 2017*). The role of these TF in human $CD4_{CTX}$ T cell differentiation remains to be determined.

The transcriptional program and lineage fate of effector T cells are established and maintained at the epigenetic level through DNA and histone modifications (*Wilson et al., 2009*; *Araki et al., 2008*; *Sellars et al., 2015*). Hypomethylation of the *PRF1* promoter, the gene encoding perforin, is associated with increased perforin expression in human CD4 T cells (*Kaplan et al., 2004*). The epigenetic modifications underlying the differentiation of $CD4_{CTX}$ T cells have not been determined.

Here, we studied circulating $CD4_{CTX}$ T cells isolated from the peripheral blood of cytomegalovirus-seropositive (CMV$^+$) healthy adults. Compared to mouse models of infection (*Brown et al., 2012*) or cancer (*Curran et al., 2013*), this situation allows access to a significant number of cells presenting a fully established cytotoxic functional program at steady state. Using transcriptomic and epigenomic approaches, we defined the molecular events that dictate human $CD4_{CTX}$ differentiation. We further show that the increased expression of Runx3 and T-bet and key epigenetic modifications at the *PRF1* promoter, without downregulation of ThPOK, underlie the acquisition of cytotoxic function by human Th1 lymphocytes.

## Results

### Phenotype and function of in vivo differentiated perforin$^+$ human CD4 T cells

In healthy humans, chronic CMV infection is associated with the expansion of perforin$^+$/granzyme B$^+$ CD4 (*Figure 1a–b* and *Figure 1—source data 1*) (*van Leeuwen et al., 2004*). In order to use this model for transcriptomic and epigenomic analyses of $CD4_{CTX}$ T lymphocytes, we characterized their phenotype and function in CMV$^+$ healthy adults and compared them to cytotoxic CD8 T cells. As previously reported, high perforin expression was observed in terminally differentiated CD4 and CD8 T cells that had downregulated the co-stimulatory molecules CD28 and CD27, respectively (*Figure 1c*) (*van de Berg et al., 2008*; *Appay et al., 2002b*). Perforin$^+$ CD4 T cells were CD8β-negative and a minority expressed low levels of CD8α (*Figure 1—figure supplement 1a–b*). Further analyses were conducted on sorted naive (CD45RO$^-$CD28$^+$) and terminally differentiated (CD28$^-$) CD4 T cells and on naive (CD45RO$^-$CD27$^+$) and terminally differentiated (CD27$^-$) CD8 T cells (*Figure 1d*). Increased *PRF1* gene expression by CD28$^-$ CD4 and CD27$^-$ CD8 T cells was confirmed by mRNA quantification and was associated with potent cytotoxic activity in a polyclonal cell lysis assay (*Figure 1e*, *Figure 1f* and *Figure 1—source data 1*). This activity was abolished by Concanamycin A, supporting a perforin-dependent mechanism (*Kataoka et al., 1996*). Bisulphite sequencing indicated an inverse correlation between the expression of the *PRF1* gene and the DNA methylation

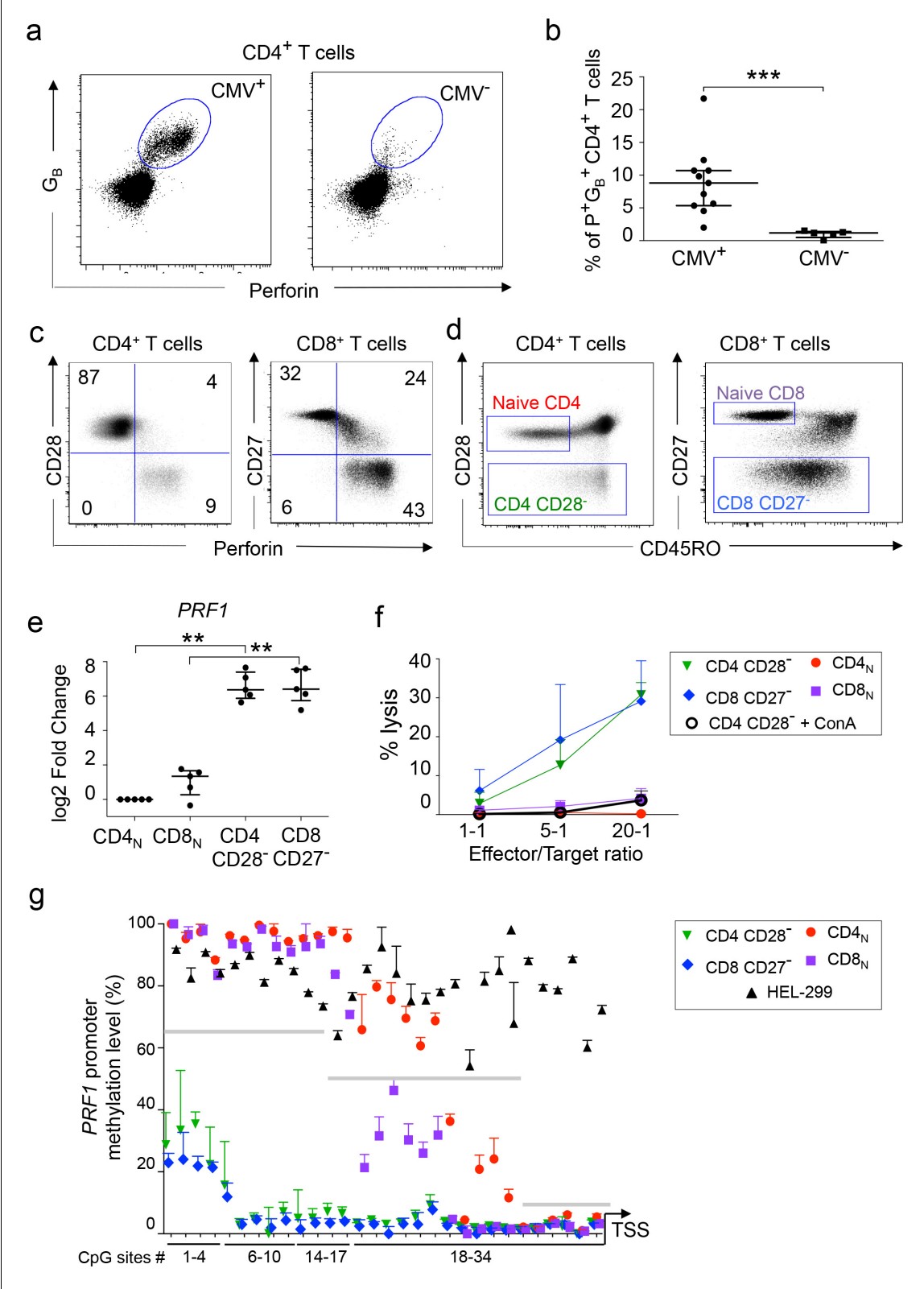

**Figure 1.** Phenotype and function of in vivo differentiated perforin+human CD4 T cells. (a–b). The expression of perforin (P) and granzyme B ($G_B$) was analyzed by flow cytometry in total CD4 T cells of CMV seropositive (CMV+) and seronegative (CMV-) healthy adults. (a) Representative dot plots of log10 fluorescence. (b) Proportions (median ±interquartile range) of P+$G_B$+ CD4 T cells in 11 CMV+ and 5 CMV- subjects. (c) Perforin expression in CD28- CD4 T cells and CD27- CD8 T cells of CMV+ subjects Numbers indicate cell proportions in each quadrant. (d) Sorting strategy of naive and

*Figure 1 continued on next page*

*Figure 1 continued*

cytotoxic T cells according to expression of CD45RO, CD28 and CD27. Representative dot plot of log10 fluorescence. (e) The expression of *PRF1* mRNA was measured by qPCR in purified T cell subsets of 5 CMV⁺ subjects. Results are median ± interquartile range of log2 fold change as compared to naive CD4 T cells. **:p<0,01 and ***:p<0,01. (f) The cytolytic activity of purified T cell subsets against anti-CD3-loaded target cells was assessed with or without pre-incubation with Concanamycin A (ConA). Data are mean ± SEM of three independent experiments on cells from different donors. (f) The methylation status of the *PRF1* promoter was assessed in T cell subsets by bisulphite pyrosequencing. Data are median ± interquartile range of five donors for CD28⁻CD4 T cells and HEL-299 and of 9 donors for the other indicated subsets. Grey lines indicate three regions with distinct methylation profiles. See also *Figure 1—figure supplement 1* and Source data file.

DOI: https://doi.org/10.7554/eLife.30496.002

The following source data and figure supplement are available for figure 1:

**Source data 1.** Phenotype and function of in vivo differentiated perforin +human CD4 T cells.

DOI: https://doi.org/10.7554/eLife.30496.004

**Figure supplement 1.** Expression of CD8 subunits by CD4$_{CTX}$ T cells.

DOI: https://doi.org/10.7554/eLife.30496.003

status of its promoter region (*Figure 1g* and *Figure 1—source data 1*). Whereas *PRF1* promoter was hypermethylated in a perforin⁻ fibroblastic cell line (HEL-299), all CpG sites were hypomethylated in CD28⁻ CD4 and CD27⁻ CD8 T cells. Low DNA methylation levels were detected at intermediate (16 to 28; middle grey line on *Figure 1g*) CpG sites in naive CD8 T cells and at proximal (CpG sites 29 to 34; right grey line) sites in both naive CD4 and CD8 T cells, suggesting that the *PRF1* gene is transcriptionally poised in naive T lymphocytes. Together, these results indicate that CD28⁻ CD4 T cells exert a cytotoxic activity comparable to CD27⁻ CD8 T cells and that this subset can therefore be used as a relevant model of in vivo differentiated CD4$_{CTX}$ T cells.

## The transcriptional program of CD4$_{CTX}$ T cells is enriched in CD8 T cell lineage genes without down regulation of ThPOK

In order to elucidate the molecular basis of CD4$_{CTX}$ T cell differentiation, their transcriptome was first compared to that of naive CD4, naive CD8 and CD8$_{CTX}$ T cells. Unsupervised analysis of transcriptional programs indicated that naive and cytotoxic T cells formed separate clusters and that CD4$_{CTX}$ and CD8$_{CTX}$ T cells were more closely related than their naive counterparts (*Figure 2a–b*). Gene set enrichment analysis (GSEA) was used to quantify the degree of sharing of the transcriptional program of CD4$_{CTX}$ T cells with the CD4 and CD8 T cell lineages. Genes expressed at higher levels in CD4$_{CTX}$ and CD8$_{CTX}$ T cells as compared to their naive counterparts were identified and their enrichment in naive CD8 and CD4 T cell transcriptomes was assessed (*Supplementary file 1* and *Figure 2c*). As expected, genes that were upregulated in CD8$_{CTX}$ T cells were significantly enriched in genes of the CD8 T cell lineage (*Figure 2c*). Strikingly, genes upregulated in CD4$_{CTX}$ T cells were also enriched in CD8 rather than CD4 T cell lineage genes. The transcriptional program common to CD4$_{CTX}$ and CD8$_{CTX}$ T cells included *RUNX3*, *TBX21* (T-bet) and *EOMES*, TF known to promote effector and memory functions in CD8 T cells (*Supplementary file 1*, *Figure 2d* and *Figure 2—source data 1*) (*Cruz-Guilloty et al., 2009*).

Strikingly, the enrichment in CD8 T cell lineage genes by CD4$_{CTX}$ T cells was not associated with the downregulation of *ZBTB7B* (ThPOK) (*Figure 2d* and *Figure 2—source data 1*). As expected, ThPOK expression was higher in naive CD4 as compared to naive CD8 T cells. However, in contrast to mouse intestinal CD4 T cells (*Mucida et al., 2013*), ThPOK gene expression was not downregulated by human circulating CD4 T cells, CD4$_{CTX}$ T cells expressing higher levels of ThPOK mRNA than naive CD4 T cells (*Figure 2d* and *Figure 2—source data 1*). Protein expression analysis by flow cytometry confirmed mRNA expression profiles for Runx3, T-bet and Eomes (*Figure 2e* and *Figure 2—source data 1*). This analysis showed high and similar expression of ThPOK in CD4$_{CTX}$ and naive CD4 T cells and a higher expression of ThPOK in CD8$_{CTX}$ as compared to naive CD8 T cells (*Figure 2e* and *Figure 2—source data 1*).

We explored the epigenetic basis of the transcriptional program of CD4$_{CTX}$ T cells by analysing their DNA methylome and comparing it to that of naive CD4 and CD8$_{CTX}$ T cells. The acquisition of cytotoxic function by CD4$_{CTX}$ and CD8$_{CTX}$ T cells was associated with changes in methylation, primarily hypomethylation, of large numbers of genes (*Figure 2—figure supplement 1a–d*). Unsupervised analysis of DNA methylomes indicated that naive and cytotoxic T cells formed separate

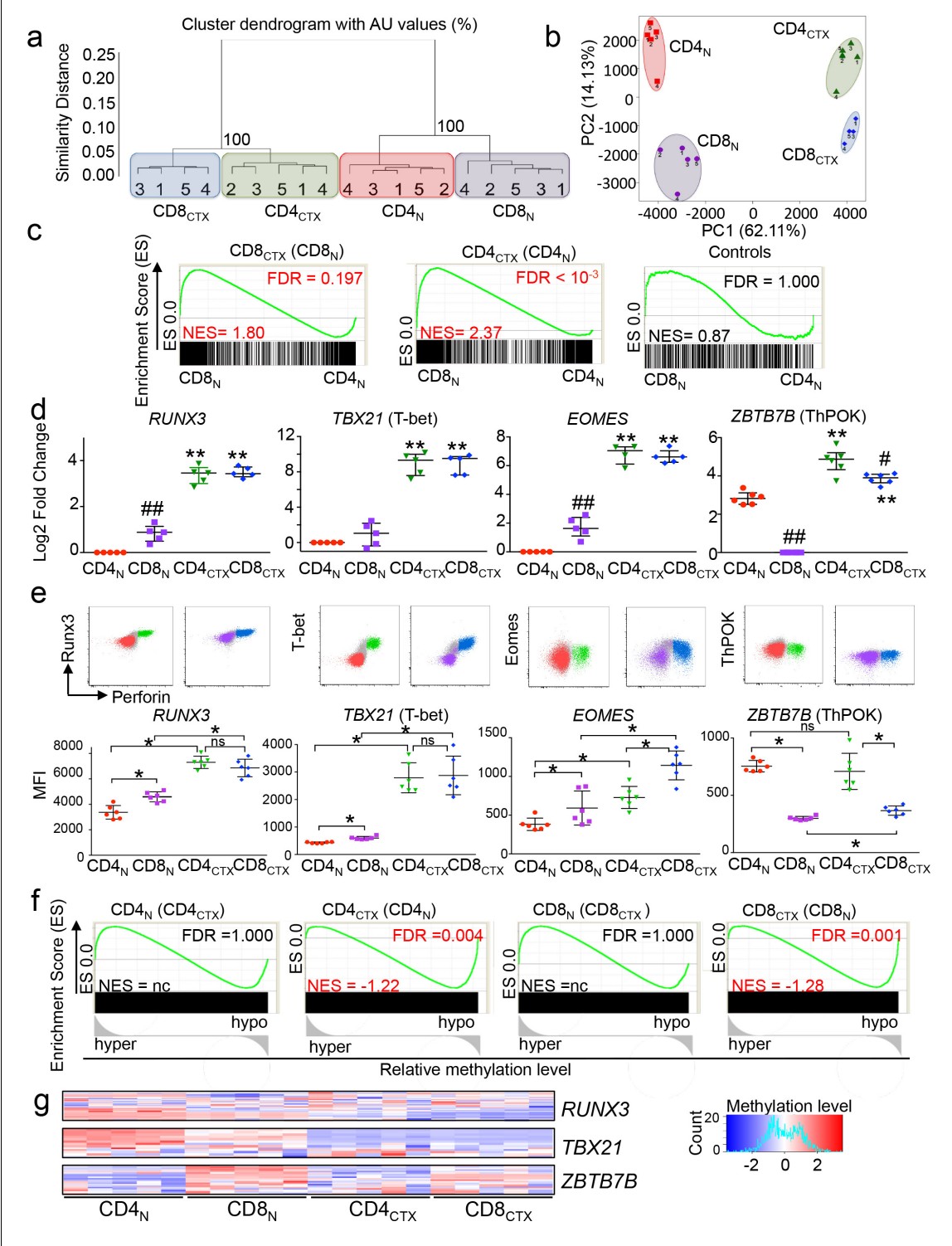

**Figure 2.** The transcriptional program of CD4$_{CTX}$ T cells is enriched in CD8 lineage genes without downregulation of ThPOK. The transcriptome of T cell subsets from five CMV-seropositive donors (four for CD8$_{CTX}$ T cells) was analyzed by gene expression arrays. Log2 expression values of 14,372 probes with a variance >0.01 corresponding to 11,200 unique genes were submitted to unsupervised clustering (**a**) and principal component (**b**) analyses. (**c**) GSEA was used to test the enrichment of CD8$_{CTX}$ and CD4$_{CTX}$ T cell GeneSets (*Supplementary file 1*) in naive CD8 and naive CD4 T cell expression datasets. Genes showing no differential expression in CD8$_{CTX}$ and CD4$_{CTX}$ T cells were used as negative controls (n = 379). Bar codes show the ranking of the log2 fold change of gene expression values in naive CD8 versus naive CD4 T cells. Green lines represent enrichment profiles. False discovery rates (FDR) below 0.25 were considered significant and are indicated in red. NES: normalized enrichment score. (**d**) TF mRNA expression by T

*Figure 2 continued on next page*

*Figure 2 continued*

cell subsets purified from four to six donors, as indicated, was assessed by qPCR (upper panels). Results are expressed as median ±interquartile range of the log2 fold change as compared to naive CD4 or naive CD8 T cells. #:p<0.05 and ##:p<0.01 as compared to CD4 T cell counterparts; *:p<0.05 and **:p<0.01 as compared to naive counterparts. (e) TF protein expression was analyzed in T cell subsets by flow cytometry Upper panels: co-expression with perforin from one representative subject. Gated populations include naive CD4 (red), CD4$_{CTX}$ (green), naive CD8 (purple) and CD8$_{CTX}$ (blue) T cells. Lower panels: individual median intensity of fluorescence (MFI) of 5 CMV$^+$ subjects. *:p<0.05. NS: not significant. Naive and cytotoxic T cell subsets were gated using the markers and strategy illustrated in *Figure 1d*. (f) GSEA was used to determine the correlation between gene expression and DNA methylation for each indicated T cell subset. Graphs show the enrichment of indicated GeneSets (*Supplementary file 1*) in genes that were either hypo or hypermethylated at the level of the promoter in the corresponding T cell subset. Black rectangles represent saturated bar codes of ranked delta beta values of 205,783 probes between the indicated subsets. (g) Heatmap of methylation beta values of *RUNX3*, *TBX21* and *ZBTB7B* gene promoter in T cell subsets of five donors.. See also *Figure 2—figure supplement 1* and Source data file.

DOI: https://doi.org/10.7554/eLife.30496.005

The following source data and figure supplement are available for figure 2:

**Source data 1.** The transcriptional program of CD4CTX T cells is enriched in CD8 lineage genes without down regulation of ThPOK.
DOI: https://doi.org/10.7554/eLife.30496.007
**Figure supplement 1.** Whole genome methylation profiles of naive and cytotoxic CD4 and CD8 T cells.
DOI: https://doi.org/10.7554/eLife.30496.006

clusters (*Figure 2—figure supplement 1e*). GSEA indicated that genes expressed at higher levels in CD4$_{CTX}$ or CD8$_{CTX}$ T cells as compared to their naive counterparts were significantly enriched in hypomethylated CpG located in their promoter regions (*Figure 2f*). In contrast, genes that were downregulated in CD4$_{CTX}$ or CD8$_{CTX}$ T cells were not enriched in hypermethylated CpG, suggesting distinct epigenetic mechanisms in the up- and downregulation of genes upon differentiation of cytotoxic T cells. Analysis of probes located in the promoter region of TF up regulated in cytotoxic as compared to naive T cells suggested significant hypomethylation for *RUNX3* and *TBX21* in both CD4$_{CTX}$ and CD8$_{CTX}$ (*Figure 2g* and *Figure 2—source data 1*), but not for *EOMES* (data not shown). As observed during thymopoiesis (*Vacchio and Bosselut, 2016*), *ZBTB7B* promoter was significantly hypomethylated in naive CD4 as compared to naive CD8 T cells, but no significant difference was observed between CD4$_{CTX}$ and CD8$_{CTX}$ cells (*Figure 2g* and *Figure 2—source data 1*). Most analyzed *RUNX3* probes (77%) were located in the distal promoter (TSS1500), a region suspected to be involved in the control of *RUNX3* mRNA translation (*Kim et al., 2015*), whereas most *TBX21* (77%) and *ZBTB7B* probes (76%) were located in the proximal promoter (5'UTR, 1st exon and TSS200). Together, these results indicate that the transcriptional program of human CD4$_{CTX}$ T cells is enriched in CD8 T cell lineage genes. Acquisition of this program involves extensive hypomethylation of the promoter regions of a large number of genes, including TF, and is not accompanied by ThPOK downregulation.

## Stepwise differentiation of CD4$_{CTX}$ T cells within the Th1 lymphocyte lineage

In order to decipher the molecular pathways involved in the differentiation of CD4$_{CTX}$ T cells, we measured the expression of cytotoxicity-related genes in subsets of memory CD4 T cells. The production of perforin was associated with the expression of the Th1 chemokine receptors CCR5 and, to a lower extend, CXCR3 but not with Th17 or Th2 receptors CCR6, CCR4 or CRTh2 (*Figure 3a*). (*Sallusto et al., 1998*; *Cosmi et al., 2000*; *Rivino et al., 2004*; *Cohen et al., 2011*; *Couturier et al., 2014*). To determine at which stage of their differentiation Th1 cells initiate the production of perforin, central memory (CM) and CD28$^+$effector memory (EM) T cells expressing Th1 chemokine receptors were compared to naive and CD4$_{CTX}$ T cells (*Figure 3b*). No perforin$^+$ cells were detected by flow cytometry among CM CD4 T cells. A small proportion of perforin$^+$ cells were detected among CD28$^+$ EM CD4 T cells in some donors and these cells were CCR5$^+$. Mean fluorescence intensity analysis indicated high perforin expression by CD4$_{CTX}$ cells and low and comparable expression in naive, CCR5$^+$ CM (CM$_{Th1}$) and CCR5$^+$ CD28$^+$ EM (EM28$^+_{Th1}$) CD4 T cells (*Figure 3c* and *Figure 3—source data 1*). In contrast, *PRF1* mRNA expression analysis of sorted cells (sorting strategy shown in *Figure 3—figure supplement 1a*) indicated that the *PRF1* gene was already expressed at higher levels in CM$_{Th1}$ as compared to naive cells and was further upregulated in EM28$^+_{Th1}$ cells and in CD4$_{CTX}$ cells (*Figure 3d* and *Figure 3—source data 1*). Notably, this pattern

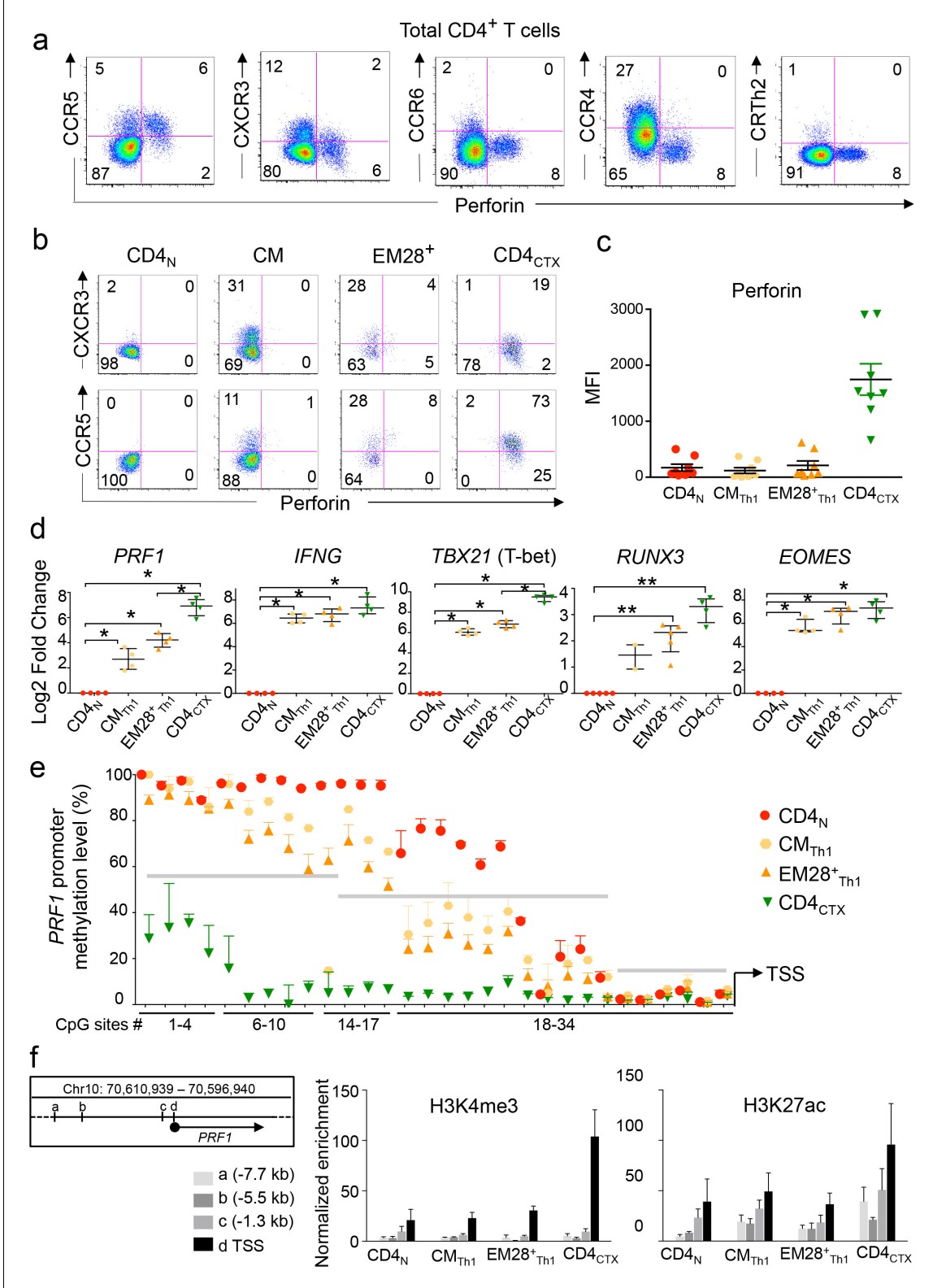

**Figure 3.** Differentiation of CD4_CTX T cells within the Th1 lineage. (**a–b**). Expression of perforin and chemokine receptors was assessed by flow cytometry in total CD4 T cells (**a**) central memory (CM), effector memory (EM) CD28[+] and CD4_CTX T cells (**b**). Log10 fluorescence of one representative donor out of 9. (**c**) Median intensity of fluorescence (MFI) of perforin expression in T cell subsets from 7 CMV[+] subjects. (**d**) mRNA expression of indicated genes was assessed by qPCR in T cell subsets purified from two to five donors, as indicated. CM_Th1 and EM28[+]_Th1 were CCR5[+]CCR6[-] as

*Figure 3 continued on next page*

*Figure 3 continued*

illustrated in Supplementary *Figure 3*. Results are median ±interquartile range of the log2 fold change as compared to naive CD4 T cells. *:p<0.05 and **:p<0.01. (e) Methylation status of *PRF1* promoter was assessed by bisulphite pyrosequencing in indicated purified T cell subsets. Results are presented as median ±interquartile range of two to five donors depending on the CpG site, as detailed in *Supplementary file 2a*. Grey lines indicate three regions with distinct methylation profiles. (e) Histone modifications at indicated regions of the *PRF1* locus were studied by ChIP-qPCR in purified CD4 T cell subsets of three donors. Results are % of input after normalization for the enrichment in pan H3. Letters refer to indicated distances from the transcription start site (TSS) (left panel). See also *Figure 3—figure supplements 1* and *2* and Source data file.

DOI: https://doi.org/10.7554/eLife.30496.008

The following source data and figure supplements are available for figure 3:

**Source data 1.** Differentiation of CD4CTX T cells within the Th1 lineage.

DOI: https://doi.org/10.7554/eLife.30496.011

**Figure supplement 1.** Sorting strategy of memory CD4 T cell subsets and expression of candidate genes.

DOI: https://doi.org/10.7554/eLife.30496.009

**Figure supplement 2.** Transcriptional program underlying the expression of perforin in Th1 cells.

DOI: https://doi.org/10.7554/eLife.30496.010

of progressive up regulation of gene expression was similar for *RUNX3* and *TBX21* and contrasted with the high expression of *IFNG* and *EOMES* mRNA already detected at the CM$_{Th1}$ stage of differentiation (*Figure 3d*, and *Figure 3—source data 1* and *Supplementary file 1*). Analysis of sorted CM Th1, Th2 and Th17 cells indicated that the increased expression of perforin and associated TF was specific to the Th1 lineage (*Figure 3—figure supplement 1b–c*). Epigenetic analyses of the *PRF1* promoter further supported the stepwise acquisition of perforin expression within the Th1 lineage. DNA methylation levels at intermediate CpGs (sites 16 to 28) progressively decreased from naive to CM$_{Th1}$, EM28$^+$$_{Th1}$ and CD4$_{CTX}$ T cells (*Figure 3e* and *Supplementary file 2a*). As observed in naive and CD4$_{CTX}$ T cells, proximal CpGs (sites 29 to 34) were hypomethylated in CM$_{Th1}$ and CD28$^+$ EM$_{Th1}$ T cells. H3K4me3 (active promoter) and H3K27ac (active promoter and enhancer [*Shlyueva et al., 2014*]) enrichment was analyzed in previously identified putative regulatory regions (*Figure 3f*, left panel) (*Pipkin et al., 2010*; *Adams et al., 2012*). In agreement with *PRF1* gene expression and promoter methylation, high enrichment of H3K4me3 and of H3K27ac was detected at the *PRF1* proximal promoter region in CD4$_{CTX}$ T cells (*Figure 3f*, middle and right panels and *Figure 3—source data 1*). Interestingly, an H3K27ac enrichment was observed in a region located 5,5 to 7,7 kb upstream of the TSS in Th1 cell subsets, suggesting the presence of an active enhancer.

Together, these results indicate the progressive acquisition of *PRF1* gene expression from CM$_{Th1}$ to EM$_{Th1}$ lymphocytes. Single-cell PCR analysis revealed that this process is related to a progressive increase in the proportion of *PRF1* mRNA$^+$ Th1 cells (*Figure 4a* and *Figure 4—source data 1*). Gene co-expression analysis at the single-cell level indicated that *PRF1* mRNA was co-expressed with distinct sets of TF in Th1 cell subsets (*Figure 4b* and *Figure 4—source data 1*). In CM$_{Th1}$ cells, *PRF1* expression was co-expressed with a relatively restricted set of TF, including *PRDM1*, *RUNX3* and *EOMES*. In CD4$_{CTX}$ T cells, a larger set of co-expressed TF was identified, including *TBX21*, *HOPX*, *ZNF683* (Hobit), *PRDM1* and *RUNX3* and *EOMES*. Notably, a lower proportion of perforin$^+$ CD4$_{CTX}$ T cells co-expressed *EOMES* as compared to the other co-expressed TF. In conclusion, the analysis of *PRF1* gene expression in vivo suggests a model in which Th1 lymphocytes acquire permissive modifications of the local chromatin environment and a network of TF factors that could underlie the acquisition of cytotoxic functions.

## Transcriptional program underlying the expression of perforin in Th1 cells

In order to identify the key steps that drive the acquisition of cytotoxic functions along the Th1 pathway, we compared the gene expression profile of CD4$_{CTX}$ T cells to that of CM$_{Th1}$ cells. Unsupervised gene expression analysis indicated that CM$_{Th1}$ cells formed a cluster separated from naive and CD4$_{CTX}$ T cells and were more closely related to CD4$_{CTX}$ T cells than to naive cells (*Figure 5a*). Including CD8 T cell subsets in this analysis indicated that the transcriptome of CM$_{Th1}$ cells was more distant to CD8$_{CTX}$ T cells than CD4$_{CTX}$ T cells (*Figure 3—figure supplement 2a–b*). 693 genes were significantly upregulated in CM$_{Th1}$ as compared to naive cells and 322 genes were up regulated in CD4$_{CTX}$ T cells as compared to CM$_{Th1}$ cells (*Figure 5b* and *Supplementary file 1*). Similar

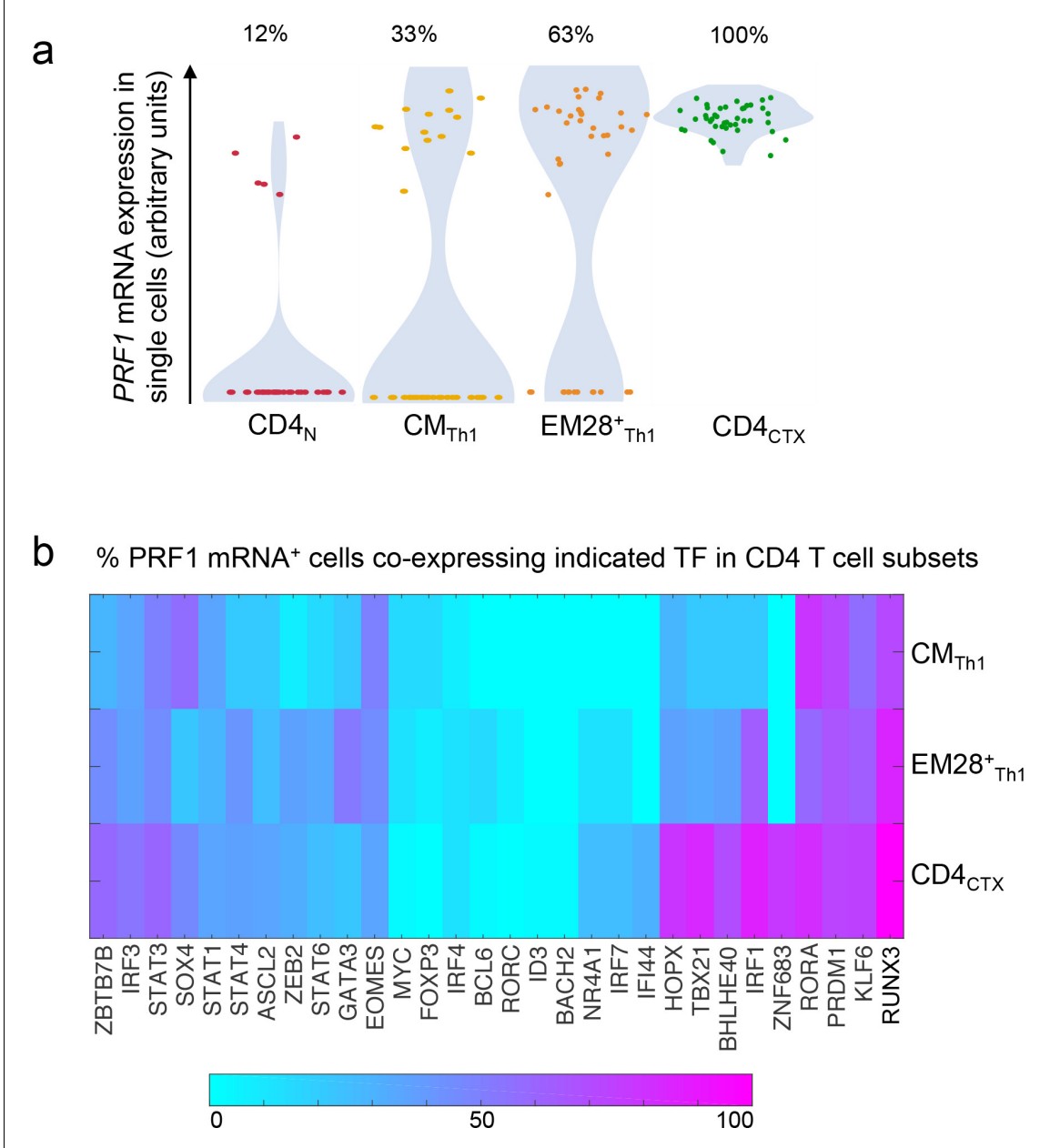

**Figure 4.** Proportion of *PRF1* mRNA[+]cells and co-expression with transcription factors in the Th1 lineage. The expression of *PRF1* and transcription factors (TF) mRNA was analyzed in 43 single naive, central memory (CM) Th1, effector memory (EM) CD28[+] Th1 and CD4$_{CTX}$ T cells from one donor. (a) *PRF1* mRNA expression in single cells and proportions of *PRF1*[+] cells in Th1 cell subsets. (b) Heat map of *PRF1*[+] cells co-expressing individual TF in Th1 cell subsets. (calculated proportion: double-positive cells/perforin-positive cells). See also *Figure 4—source data 1*.

DOI: https://doi.org/10.7554/eLife.30496.012

The following source data is available for figure 4:

**Source data 1.** Proportion of PRF1 mRNA +cells and co-expression with transcription factors in the Th1 lineage.

DOI: https://doi.org/10.7554/eLife.30496.013

gene numbers were downregulated in the two subsets. As expected, genes upregulated in CD4$_{CTX}$ T cells as compared to CM$_{Th1}$ cells included cytotoxicity-related molecules, among which *GNLY* (granulysin), granzymes, CD107a (*LAMP1*), *CX3CR1* and *CD8A* as well as TF *RUNX3* and *EOMES* (*Figure 5c–d*). T-bet was not up regulated in the transcriptome dataset, in line with reported lack of sensitivity of the Illumina array for this gene (*Dimova et al., 2015*). Expression of *IFNG*, *TNF* and

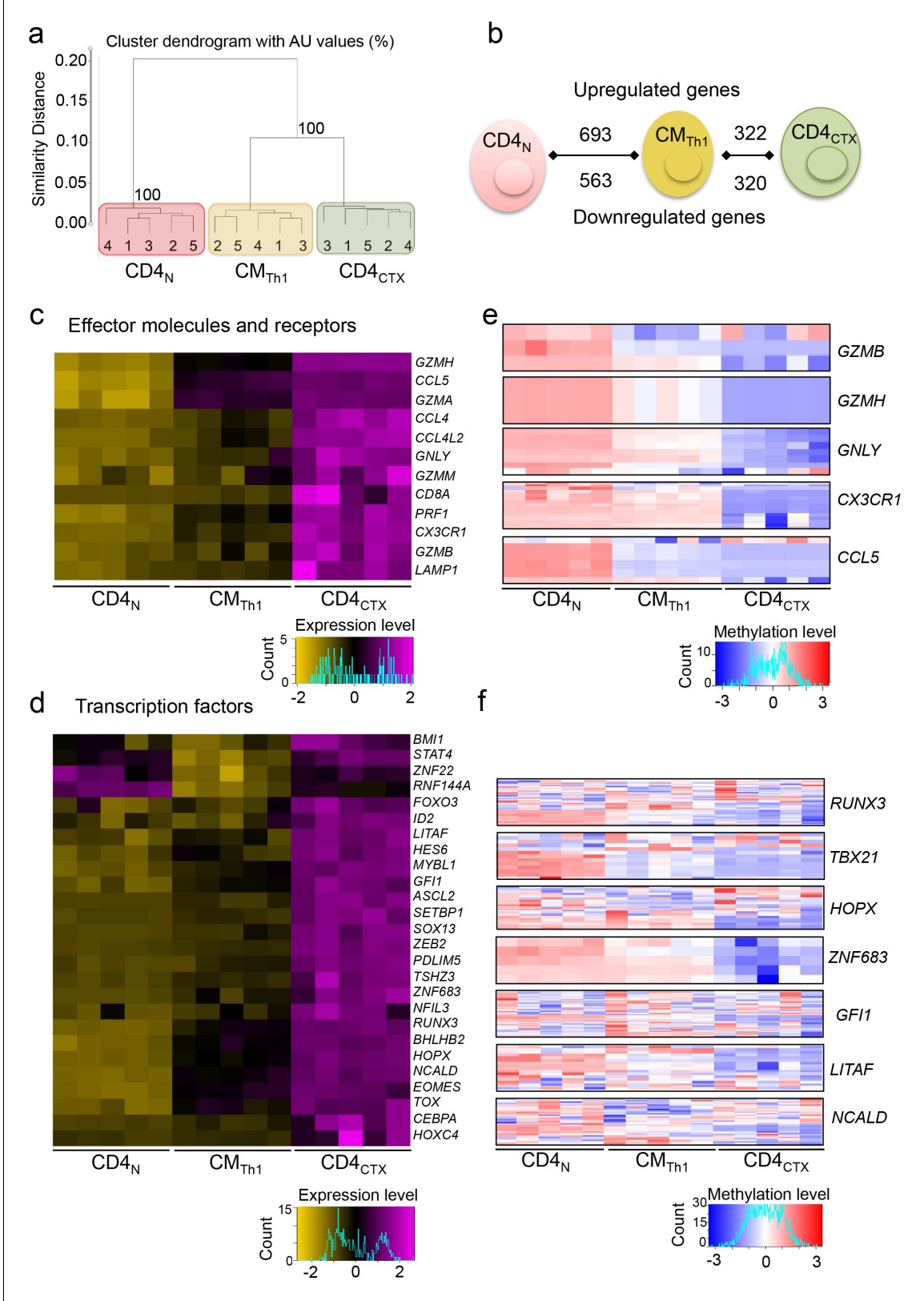

**Figure 5.** Transcriptional program underlying the expression of perforin in Th1 cells. The transcriptome of CD4 T cell subsets from five donors was analyzed by gene expression arrays. (a) Log2 expression values of 13,551 probes with a variance >0.01 corresponding to 10,669 unique genes were submitted to unsupervised cluster analysis. See also Supplementary *Figure 4*. (b) Genes differentially expressed by naive CD4 (CD4$_N$), central memory

*Figure 5 continued on next page*

*Figure 5 continued*

(CM) Th1 and CD4$_{CTX}$ T cells were identified. (c-d) Heatmaps of mean log2 expression values of all probes for each selected effector molecules and receptors (c) and transcription factors (d) upregulated in CD4$_{CTX}$ T cells as compared to CM$_{Th1}$ cells. See also *Figure 5—figure supplement 1*. (e-f) Heatmaps of methylation beta values of all the probes located in the promoter region of effector molecules and receptors (e) and transcription factors (f) significantly hypomethylated in CD4$_{CTX}$ as compared to CM$_{Th1}$ as assessed by a two-way ANOVA with Tukey's multiple comparisons (p<0.01). See also *Figure 5—figure supplement 2* and Source data file.

DOI: https://doi.org/10.7554/eLife.30496.014

The following source data and figure supplements are available for figure 5:

**Source data 1.** Transcriptional program underlying the expression of perforin in Th1 cells.

DOI: https://doi.org/10.7554/eLife.30496.017

**Figure supplement 1.** Transcriptional program of CM$_{Th1}$ and CD4$_{CTX}$ T cells.

DOI: https://doi.org/10.7554/eLife.30496.015

**Figure supplement 2.** Epigenetic and transcriptional program underlying the expression of perforin in Th1 cells.

DOI: https://doi.org/10.7554/eLife.30496.016

*GZMK* was increased in CM$_{Th1}$ as compared to naive cells but was not further up regulated in CD4$_{CTX}$ T cells (*Supplementary file 1*).

Additional TF were upregulated in CD4$_{CTX}$ T cells (*Figure 5d*). Most of them were selectively upregulated in CD4$_{CTX}$ T cells as compared to CM$_{Th1}$ cells or were up regulated from naive T cells to CM$_{Th1}$ cells and from CM$_{Th1}$ cells to CD4$_{CTX}$ T cells. Some TF were first downregulated in CM$_{Th1}$ as compared to naive cells and were up regulated in CD4$_{CTX}$ T cells as compared to CM$_{Th1}$ cells, indicating a more complex regulation pathway (*Figure 5—figure supplement 1* and *Supplementary file 1*). Methylome analysis indicated that gene up regulation in CM$_{Th1}$ and CD4$_{CTX}$ T cells was primarily associated with DNA hypomethylation, as observed in the naive to CD4$_{CTX}$ T cell differentiation (*Figure 5—figure supplement 2a*). Analysis of probes located in the promoter region of effector molecules and receptors and of TF upregulated in CD4$_{CTX}$ cells were also hypo-methylated in these cells (*Figure 5e–f* and *Figure 5—source data 1*).

Among the upregulated TF identified through transcriptomic analysis, *ASCL2*, *HOPX*, *ZEB2* and *ZNF683* (Hobit) were selected for further analyses because of the high magnitude of their expression in CD4$_{CTX}$ T cells (*Figure 5—figure supplement 2b* and *Supplementary file 1*) and because they have been previously studied for their effector functions in T cells (*Liu et al., 2014*; *Albrecht et al., 2010*; *Dominguez et al., 2015*; *van Gisbergen et al., 2012*; *Vieira Braga et al., 2015a2015a*).

## TF controlling the expression of perforin in CD4 T cells

The role of selected TF in the acquisition of cytotoxic function by CD4 T cells was analyzed in a novel in vitro model of CD4$_{CTX}$ T cell differentiation involving the stimulation of naive CD4 T cells in the presence of Th1, Th2 or Th17 polarizing cytokines and the analysis of perforin and granzyme B expression at multiple time points (*Figure 6a*). In line with our in vivo observations, cytotoxicity was acquired under Th1 and not under Th2 or Th17 culture conditions (*Figure 6b* to e and *Figure 6—source data 1*). In vitro differentiated Th1 cells included a diverse repertoire of clonotypes with similar proportions of large and intermediate expansions as compared to Th1 and Th2 cells (*Figure 6—figure supplement 1a*). Furthermore, Th1 cells expressed low levels of the *PLZF* TF (*Figure 6—figure supplement 1b*) indicating that they were conventional and not innate-type effector T lymphocytes.

Frequencies of perforin$^+$granzyme B$^+$ cells and levels of *PRF1* mRNA increased from day 7 to day 21 in Th1 polarized cells (*Figure 6b–c–e* and *Figure 6—source data 1*). *GNLY* and *TNF* mRNA levels increased following a similar kinetics, whereas *GZMB*, *IFNG* and *GZMK* already reached maximum levels at day 7 (*Figure 6e*, *Figure 6—figure supplement 2*, and *Figure 6—source data 1*). *CX3CR1* mRNA presented a unique expression profile with an initial upregulation followed by a marked downregulation (*Figure 6—figure supplement 2a*). At day 14, Th1 polarized cells acquired potent Concanamycin A-sensitive cytotoxic activity (*Figure 6d* and *Figure 6—source data 1*).

Most selected TF were induced by cell activation but their pattern of expression was differently associated with polarizing conditions (*Figure 6e* and *Figure 6—source data 1*). *TBX21* and *HOPX* expression was specifically induced under Th1 conditions. *RUNX3* and *ZBTB7B* mRNA were upregulated in all conditions but reached higher levels in Th1 cells. *EOMES* was induced in both Th1 and

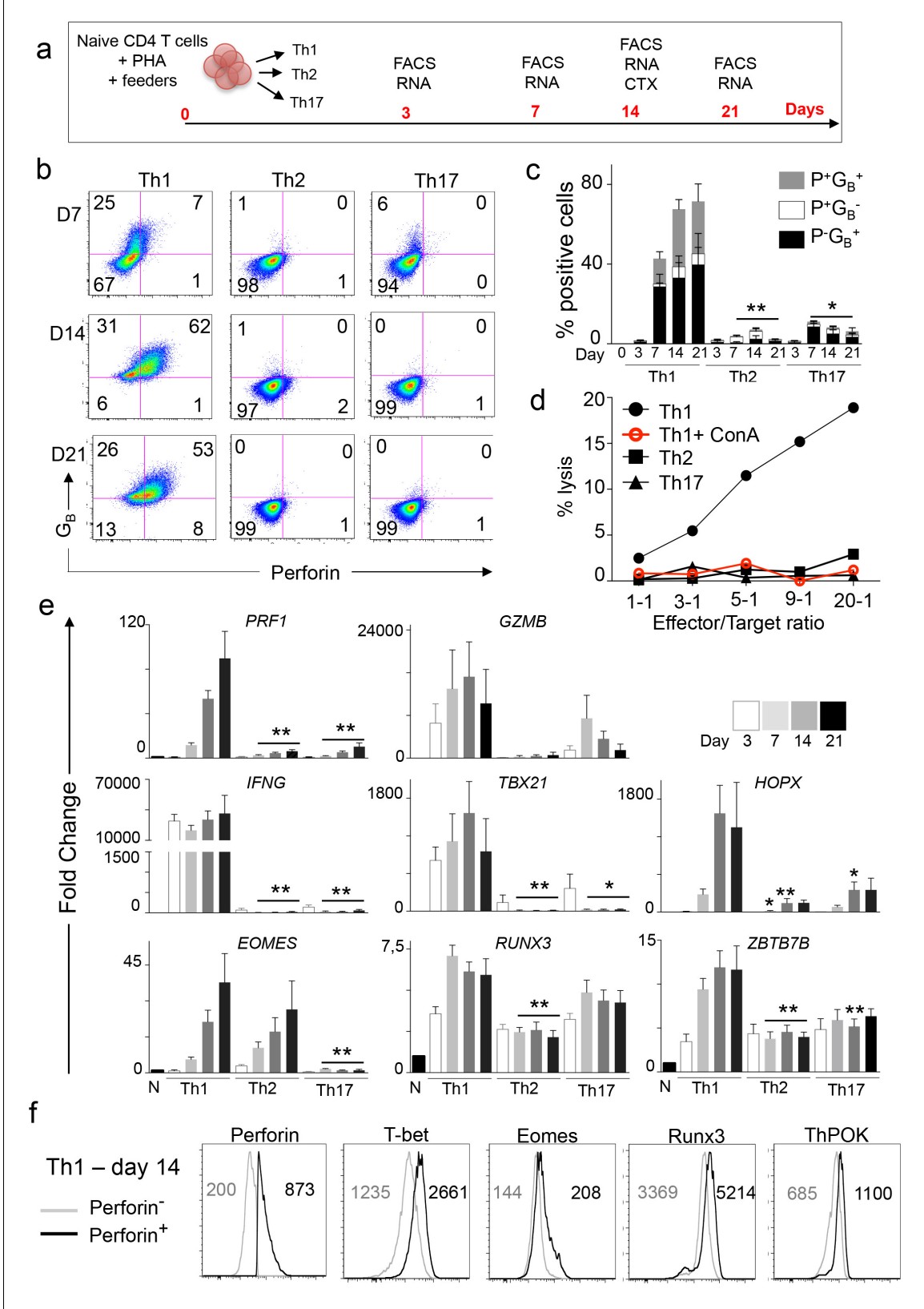

**Figure 6.** Naive CD4 T cells differentiate into CD4_CTX in Th1 culture conditions in vitro. (**a**) Naive CD4 T cells were stimulated polyclonally in the presence of Th1, Th2 or Th17 polarizing cytokines. Flow cytometry (FACS), gene expression (RNA) and cytotoxicity analyses were performed at the indicated time points. (**b-c**) Perforin (P) and GranzymeB (G_B) expression was assessed by flow cytometry. (**b**) Dot plots (Log10 fluorescence of one

*Figure 6 continued*

representative out of six different donors. Numbers indicate % of cells in individual quadrants. (**c**) Mean ±SEM of % of perforin (P) and granzyme B (G$_B$) positive cells from six independent experiments on different donors (only 4 for day 21 and 5 for day 3). (**d**) Cytotoxic activity of in vitro differentiated effector T cells against anti-CD3-loaded target cells was assessed at indicated effector/target ratios with or without pre-incubation with Concanamycin A. Figure shows one representative out of four experiments on different donors. (**e**) mRNA expression of indicated genes was quantified by qPCR. Results are mean ±SEM fold change as compared to naive CD4 T cells from six independent experiments on different donors (only 3 for day 3). *: p<0.05 and **:p<0.01 as compared to Th1 condition at the corresponding time point. (**f**) Expression of TF was measured by flow cytometry in perforin$^{high}$ and perforin$^{low}$ Th1 cells after 14 days of in vitro stimulation. Histograms of one representative out of six experiments. Numbers are median fluorescence intensity (MFI) of six experiments (only four for ThPOK). See also *Figure 6—figure supplements 1* and *2* and Source data file.

DOI: https://doi.org/10.7554/eLife.30496.018

The following source data and figure supplements are available for figure 6:

**Source data 1.** Naive CD4 T cells differentiate into CD4$_{CTX}$ in Th1 culture conditions in vitro.
DOI: https://doi.org/10.7554/eLife.30496.021
**Figure supplement 1.** TCR repertoire of and PLZF expression by in vitro differentiated CD4 T cells.
DOI: https://doi.org/10.7554/eLife.30496.019
**Figure supplement 2.** Expression of TF and effector molecules by in vitro differentiated CD4 T cells.
DOI: https://doi.org/10.7554/eLife.30496.020

Th2 cells, suggesting that expression of this TF is not sufficient to promote cytotoxic function in CD4 T cells. The pattern of expression of *ASCL2* and *ZEB2* was inconsistent and *ZNF683* mRNA was not detected in in vitro differentiated effector T cells (*Figure 6—figure supplement 2a*). Flow cytometry analysis of Th1 cells indicated that the expression of perforin was correlated with those of T-bet, Eomes, Runx3 and ThPOK (*Figure 6f* and *Figure 6—figure supplement 2b*).

Based on these results, we further evaluated the role of Runx3, T-bet, Eomes, Hopx and ThPOK in this model using shRNA silencing (*Figure 7a*). Significant knockdown was achieved for each target TF at mRNA and protein levels (*Figure 7—figure supplement 1a*), whereas the non-silencing (N-S) shRNA and the empty vector (EV) had no significant effect on any of the studied genes (*Figure 7—figure supplement 1b–c*). Knockdown of Runx3 and, to a lesser extend, T-bet resulted in the decreased expression of perforin and granzyme B. Knockdown of Hopx affected *GZMB* but not *PRF1* mRNA expression, whereas knockdown of Eomes had no significant effect on the expression of either perforin or granzyme B (*Figure 7b–d* and *Figure 7—source data 1*). Of note, *IFNG* expression was principally dependent on T-bet with no significant effect of Runx3 knockdown (*Figure 7b*). T-bet also controlled *TNF*, *CX3CR1* and *GZMK* expression whereas *GNLY* was controlled by both Runx3 and T-bet (*Figure 7—figure supplement 1b*). Significant interactions were observed between TF, with Runx3 and T-bet controlling the expression of *HOPX* and T-bet and Hopx controlling the expression of *EOMES* (*Figure 7—figure supplement 1c*). In line with its central influence on the expression of cytotoxic molecules, Runx3 also significantly modulated the acquisition of cytotoxic activity by CD4 T cells, whereas T-bet knockdown reduced cytotoxicity of CD4 T cells only at low E/T ratios (*Figure 7e* and *Figure 7—source data 1*). ThPOK globally acted as a negative regulator of the cytotoxic program (*Figure 7f–g*, *Figure 7—figure supplement 1b–c* and *Figure 7—source data 1*). ThPOK knockdown markedly increased *PRF1* and *GZMB* expression and cytotoxic activity of CD4 T cells and also upregulated the expression of all the other TF studied and of *CD8A*.

Together, these results indicate that the acquisition of a cytotoxic program by naive CD4 T cells is dependent on Runx3 and, to a lesser extend, T-bet and is limited by the sustained expression of the CD4 lineage TF ThPOK. Because ThPOK was upregulated in CD8$_{CTX}$ T cells in vivo (*Figure 2d–e* and *Figure 2—source data 1*), we studied its expression and role in the in in vitro differentiation of CD8 T cells. In vitro activation of naive CD8 T cells induced the differentiation of perforin$^+$ cells that co-expressed ThPOK (*Figure 7—figure supplement 2a*). Knockdown of ThPOK in differentiated CD8 T cells did not significantly influence the expression of perforin but significantly increased the expression of granzyme B as compared to N-S shRNA (*Figure 7—figure supplement 2b*), suggesting that ThPOK may limit the cytotoxic function of human CD8 T cells.

## Discussion

This study demonstrates that the acquisition of cytotoxic function by human CD4 T cells is an integral part of the Th1 linear differentiation pathway. Several concordant observations support this conclusion. First, *PRF1* gene expression was detected in all subsets of memory Th1 cells. The proportion of *PRF1* expressing cells increased from CM to terminally differentiated EM Th1 cells and this process was associated with the diversification of co-expressed TF networks. The epigenetic modifications detected at the *PRF1* gene promoter reflect this increase in perforin expression and suggest that the local chromatin environment becomes progressively more favorable from naive T cells to CM Th1 cells to terminally differentiated EM Th1 cells. Progressive acquisition of specific transcriptional and epigenetic marks is a hallmark of the linear differentiation model of CD4 T cell memory development (*DEEP Consortium et al., 2016*). Our work suggests that acquisition of cytotoxic functions by CD4 T cells follows a similar stepwise program.

The accessibility and expression of *PRF1* gene in all subsets of Th1 cells also provide a basis for the classical observation that memory Th1 cells of diverse antigen-specificities that do not express the perforin protein ex vivo become perforin positive and cytotoxic upon in vitro expansion and cloning (*Parronchi et al., 1992*; *Riaz et al., 2016*). This in vitro acquisition of perforin expression likely reflects pre-established *PRF1* chromatin modifications in precursors of $CD4_{CTX}$ T cells.

The cytotoxic potential of Th1 cells is also supported by the in vitro model of $CD4_{CTX}$ T cell differentiation. In this model, human naive CD4 T cells stimulated in the presence of Th1, and not Th2 or Th17, polarizing cytokines differentiated in perforin$^+$granzyme B$^+$ cells with potent cytotoxic activity. This could be due to the promotion of the cytotoxic phenotype by Th1 cytokines or to its repression by the Th2 or Th17 transcriptional programs, or both (*Parronchi et al., 1992*; *Xiong et al., 2013*; *Ciucci et al., 2017*). In contrast to the rapid acquisition of *IFNG* expression, naive CD4 T cells expressed high levels of *PRF1* 1 to 2 weeks after stimulation. This observation also contrasts with the more rapid acquisition of cytotoxic function by CD8 T lymphocytes (*Araki et al., 2008*) and suggests that the initial steps of Th1 cell differentiation provide the required epigenetic and transcriptional signals promoting the expression of *PRF1*. The relatively delayed acquisition of cytotoxicity also suggests that $CD4_{CTX}$ T cells are induced by prolonged antigen stimulation in vivo and may intervene when Th1 and CD8 T cells do not adequately control pathogens.

The acquisition of cytotoxic function within the Th1 lineage was promoted by Runx3 and T-bet. Runx3 knockdown reduced the expression of perforin, granzyme B and granulysin by $CD4_{CTX}$ and decreased their cytotoxic activity. In contrast to mouse Th1 cells, Runx3 did not influence IFN-γ mRNA expression by human Th1 cells (*Djuretic et al., 2007*; *Wang et al., 2014*). This discrepancy may be related to inter-species differences or to incompleteness of the knockdown in our experimental conditions. T-bet knockdown reduced the expression of IFN-γ and cytotoxic molecules but had a more moderate impact than Runx3 knockdown on the acquisition of cytotoxic activity. This result is in line with a recent report indicating a role of T-bet in the induction of cytotoxic molecules by TCR-engineered tumor-specific CD4 T cell lines (*Jha et al., 2015*). Together these results indicate that human Th1 cells acquire cytotoxic functions under the control of the master regulator Runx3 cooperating with T-bet (*Cruz-Guilloty et al., 2009*; *Jha et al., 2015*). Runx3 and T-bet also controlled the expression of other TF that were not directly involved in the acquisition of cytotoxic function in these experimental conditions, including Eomes and Hopx.

Eomes was upregulated in $CD4_{CTX}$ T cells as compared to naive cells in vivo but its expression was less correlated with perforin than Runx3 and T-bet. In vitro, Eomes was neither sufficient nor necessary to induce cytotoxicity as it was upregulated in Th2 cells that did not express perforin and its knockdown did not impact the expression of perforin in Th1 cells. Together, these results suggest that the role of Eomes in the acquisition of cytotoxic function by human CD4 T cells may be limited. This contrasts with a report indicating that overexpression of Eomes induces cytotoxic function in CD4 T cell lines, suggesting that this TF may promote cytotoxicity when expressed at high levels in CD4 T cells, as observed in CD8 T cells (*Pearce et al., 2003*; *Eshima et al., 2012*; *Intlekofer et al., 2008*). The role of Eomes could also be restricted to specific conditions of co-stimulation as recently reported (*Mittal et al., 2018*). The expression of Hopx by $CD4_{CTX}$ T cells showed a similar pattern as T-bet and Runx3 both in vivo and in vitro. Its knockdown reduced the expression of granzyme B but did not significantly impact perforin. Therefore, the primary role of Hopx in $CD4_{CTX}$ T cells may not be the induction of cytotoxicity but may include other functions, including survival (*Albrecht et al., 2010*).

Two other TF, Hobit (*ZNF683*) and ZEB2, were specifically expressed by $CD4_{CTX}$ T cells in vivo but not in vitro. Hobit is upregulated in human effector CD8 T cells and murine NKT cells and therefore appears to be part of a signature common to cytotoxic lymphoid cells (*van Gisbergen et al., 2012*; *Vieira Braga et al., 2015a2015a*; *Vieira Braga et al., 2015b*). A recent study revealed that Hobit induces a transcriptional program promoting tissue residency of memory T cells and suggests that it could regulate cytotoxic function in murine NKT1 cells (*Mackay et al., 2016*). Its role in the promotion of cytotoxic function by CD4 T cells in vivo therefore remains to be established. Similarly, further studies should establish the role of ZEB2 in terminally differentiated CD4$^+$ T lymphocytes (*Dominguez et al., 2015*; *Omilusik et al., 2015*).

In parallel with the positive regulation operated by T-bet and Runx3, the cytotoxic function of human CD4 T cells was negatively regulated by ThPOK. In CD4$^+$CD8$^{lo}$ thymocytes, ThPOK decreases the expression of Runx3 and perforin and promotes the development of the CD4 T cell lineage (*Liu et al., 2005*). In the periphery, ThPOK also inhibits the expression of Runx3, T-bet and Eomes by mouse CD4 T cells and restricts their cytotoxic differentiation (*Wang et al., 2008*). We observed that the expression of Runx3 and ThPOK is not mutually exclusive in human $CD4_{CTX}$ and $CD8_{CTX}$ T cells. Co-expression of Runx3 and ThPOK has been observed in murine Th1 cells (*Djuretic et al., 2007*) and in simian MHCII-restricted $CD8_{\alpha\alpha}$ T cells after CD4 downregulation (*Vinton et al., 2017*). Also, ThPOK is up regulated by mouse effector CD8 T cells during acute viral infection and promotes their expansion and effector function upon rechallenge (*Setoguchi et al., 2009*). On the other hand, ThPOK is required for the development of other murine lymphoid subsets with cytotoxic potential including CD4$^+$ NKT cells and γδ T cells (*Wang et al., 2010*; *Park et al., 2010*). Together, these observations indicate that the expression of high levels of ThPOK is compatible with the expression of Runx3 and with cytotoxic function. Yet, the in vitro model of T cell differentiation revealed that ThPOK is an important regulator of the cytotoxic activity of human CD4 and possibly CD8 T lymphocytes. This observation suggests that ThPOK may contribute to the regulation of antiviral, anticancer and immunopathological properties of Th1 cells in vivo.

In conclusion, this study shows that Runx3 and ThPOK cross-regulate the acquisition of cytotoxic function by Th1 lymphocytes and therefore represent targets for interventions against viral infections, cancer and autoimmune disorders.

# Materials and methods

**Key resources table**

| Reagent type (species) or resource | Designation | Source or reference | Identifiers |
|---|---|---|---|
| Cell line (mouse leukemia) | RAW 264.7 | ATCC Cat# TIB-71 | RRID:CVCL_0493 |
| Cell line (human kidney cell line) | HEK-293 human kidney cell line | ATCC Cat# CRL-1573, | RRID:CVCL_0045 |
| Cell line (human) | HEL 299 | ATCC Cat# CCL-137, | RRID:CVCL_2480 |
| Transfected constructs | References of all shRNAs are listed in *Supplementary file 2f* | | |
| Antibody-ChIP | References of used antibodies are indicated in the method section | | |
| Antibody-cytometry | References of all used cytometry antibodies, including company and clone are listed in *Supplementary file 2b*. | | |
| Sequence-based reagent | All sequences are listed in *Supplementary files 2c, d and e* | | |
| Chemical compound, drug | Concanamycin A | Sigma-Aldrich (Merck, Germany) | C 9705 |
| Other | PKH26 staining | Sigma-Aldrich (Merck, Germany) | Catalog numbers MINI26 and PKH26GL |

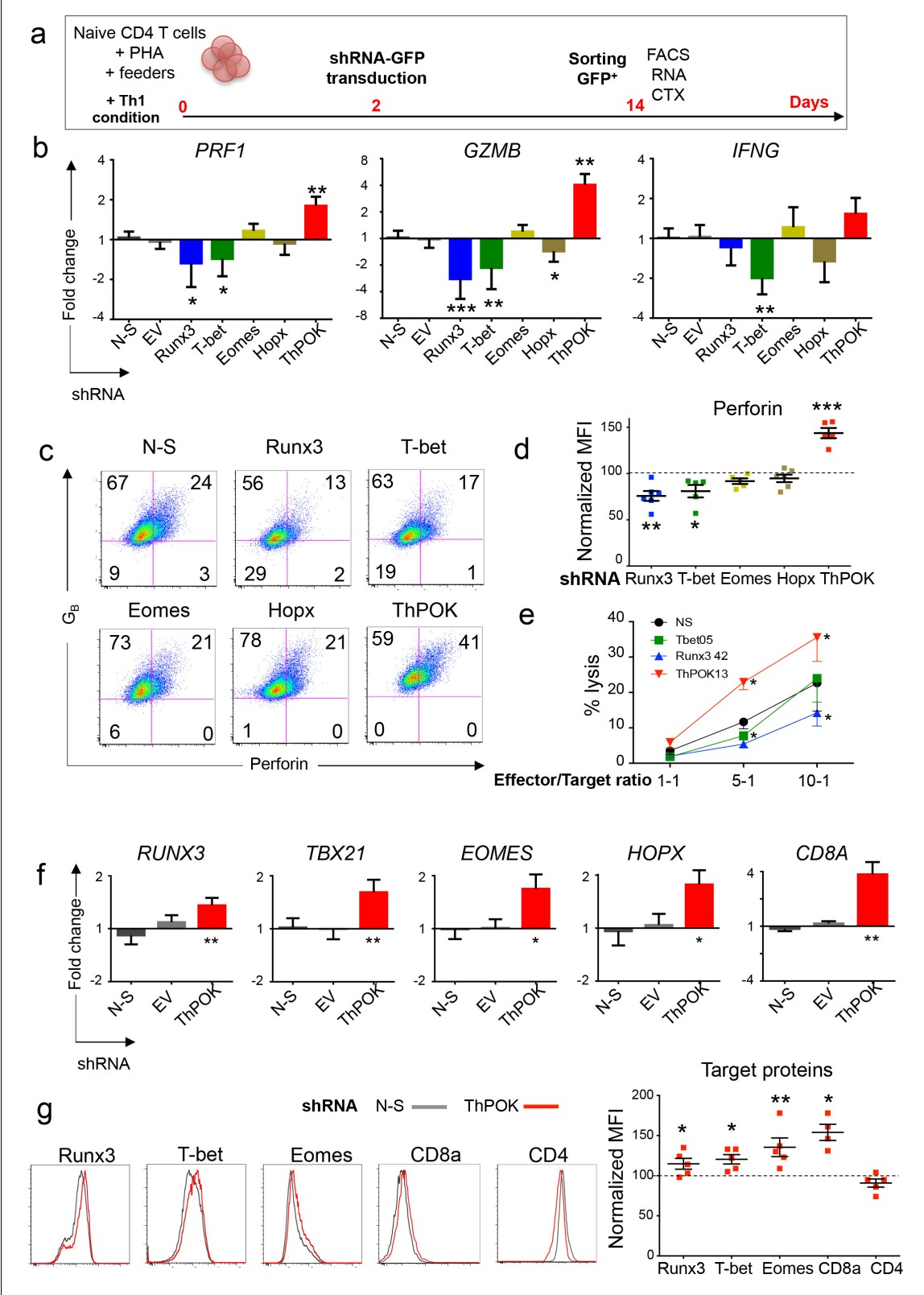

**Figure 7.** TF controlling the expression of perforin in CD4 T cells. (a) Naive CD4 T cells were stimulated polyclonally in the presence of Th1 polarizing cytokines and were transduced on day 2 with shRNA-GFP. GFP+ cells were purified on day 14 for flow cytometry, mRNA expression and cytotoxicity analyses. (b) Expression of *PRF1*, *GZMB* and *IFNG* mRNA was quantified by qPCR in cells transduced with indicated targeting shRNA, non-silencing (N-

*Figure 7 continued on next page*

*Figure 7 continued*

S) shRNA or empty vector (EV). Data are mean ± SD log2 fold change as compared to controls (mean of N-S and EV), from seven biological replicates generated in four independent experiments on different donors. (c-d) Expression of perforin (P) and granzyme B (G$_B$) was measured by flow cytometry in cells transduced with N-S or indicated gene targeting shRNA. (c) Representative dot plot (log10 fluorescence) of the proportions of double positive cells from seven independent experiments on different donors. Numbers indicate % of cells in individual quadrants. (d) Perforin expression following knockdown of indicated TF. Data are median fluorescence intensity (MFI) normalized for perforin MFI in cells transduced with N-S shRNA (dotted line, 100%). Experiments in which gene knockdown was below 10% were excluded from the analysis. (e) Cytotoxic activity of transduced cells against anti-CD3-loaded target cells. Data are mean ± SEM of five independent experiments on different donors. *:p<0.05 as compared to N-S shRNA. (f-g) The effect of ThPOK knockdown on the expression of indicated TF, CD8A and CD4 was studied by qPCR and flow cytometry. (f) Data are mean ± SD log2 fold change as compared to mean RNA expression in control shRNA (N-S and EV) from seven biological replicates generated in four independent experiments on different donors. *:p<0.05; **:p<0.01 as compared to control shRNA. (g) Histograms (Log10 fluorescence) from one representative donor. Data are MFI normalized for TF MFI in cells transduced with N-S shRNA (dotted line, 100%) of seven independent experiments on different donors. See also *Figure 7—figure supplements 1* and *2* and Source data file.

DOI: https://doi.org/10.7554/eLife.30496.022

The following source data and figure supplements are available for figure 7:

**Source data 1.** TF controlling the expression of perforin in CD4 T cells.

DOI: https://doi.org/10.7554/eLife.30496.025

**Figure supplement 1.** Knockdown of transcription factors (TF) expression by shRNA.

DOI: https://doi.org/10.7554/eLife.30496.023

**Figure supplement 2.** Influence of ThPOK on the differentiation of cytotoxic CD8 T cells.

DOI: https://doi.org/10.7554/eLife.30496.024

## Cell collection and purification

Blood samples were collected from CMV-seropositive or seronegative healthy adult volunteers, aged 26 to 60 (median (interquartile range): 45 (39.5–51.25)) years. Volunteers were recruited by the research centre ImmuneHealth, CHU Tivoli, La Louvière. Clinical staff informed the volunteers about the objectives of the study and obtained their written consent to use the human material for research purposes. The study and the informed consent form were approved by the Ethics committee of the CHU Tivoli, La Louvière, Belgium (Reference B09620097253). The study followed the Good Clinical Practice (ICH/GCP) guidelines, the Belgian Law and the declaration of Helsinki ('World Medical Association Declaration of Helsinki; Ethical Principles for Medical Research Involving Human Subjects'). Transcriptomic and methylomic analyses were conducted on 3 CMV-seropositive women, aged 38, 48 and 60 years, and 2 CMV-seropositive men, aged 40 and 52 years.

The number of samples analyzed in each experiment was defined on the basis of previous experience of the investigators or on published literature. No sample size was computed. Peripheral blood mononuclear cells (PBMC) were purified by density gradient centrifugation and stained with titrated conjugated antibodies. Cells were sorted on a BD FACS Aria III or acquired with a BD LSR Fortessa cytometer and data were analyzed using the FlowJo software (v9.2). The RAW 264.7 murine macrophage cell line (RRID:CVCL_0493; obtained from ATCC) and the HEK-293 human kidney cell line (RRID:CVCL_0045, obtained from ATCC) were cultured in DMEM (Lonza) supplemented with 10% fetal calf serum, 1% AAG, 1% Na Pyruvate and 1% Pen/strep. The HEL-299 fibroblastic cell line (RRID:CVCL_2480; obtained from ATCC) was cultured in EMEM (Lonza) supplemented with 10% fetal calf serum, 1% NEAA, 1% Hepes, 1% Glutamine, 1% Na Pyruvate and 1% Pen/strep. All cell lines were tested negative for mycoplasma infection (MycoAlert, Lonza). Because they were used only as tools to produce lentivirus particles, as targets of cytotoxic cells, and as a negative control in one methylation analysis, they were not re-authenticated after purchase.

## FACS-staining

For membrane staining, cells were washed with PBS containing 0.1% bovine serum albumin (BSA). Antibodies were incubated in PBS + 0.1% BSA for 10 min at 37°C or 15 min at room temperature. Cells were then washed with PBS + 0.1% BSA before addition of Cellfix (BD) or intracellular staining. Cells were permeabilized for intranuclear and intravesicular staining using the FoxP3 staining kit (eBiosciences). Active caspase3 staining was performed using the cytofix-cytoperm and Permwash buffers (BD). References of used antibodies are presented as *Supplementary file 2b*.

## FACS-sorting

Naive CD4 T cells were isolated by negative selection for in vitro stimulation. Before cell sorting, fresh PBMC were enriched in CD4 T cells with the Miltenyi human CD4 +T Cell Isolation Kit. Membrane staining was then performed as mentioned above with a dump channel in PE including CD14, CD19, CD16, CD56, TCRgd and CD8 mAbs. CD25-, CD45RO- and CXCR3-negative cells were further selected in order to exclude regulatory T cells, memory and stem cell memory CD4 T cells, respectively. A small fraction of these untouched naive CD4 T cells (CD3$^+$CD4$^+$CD45RO$^-$CCR7$^+$CD28$^+$) cells were then stained to verify their naive phenotype. Cell purity was 96 [94-97]% (Median [IQ]). CD4 and CD8 T cell subsets were isolated by positive selection. Before cell sorting, fresh PBMC were depleted of glycophorin A-, CD19- and CD14-positive cells as well as CD8- or CD4-positive cells using an Automacs instrument (Miltenyi). Membrane staining was then performed as mentioned above. Cells were resuspended in complete antibiotics-containing medium, sorted and collected in the same complete medium, centrifuged and lysed in RLT Plus buffer +10 µl betamercaptoethanol for later nucleic acid extraction or immediately tested for cytotoxic activity. Cell purity was 98 [95-99]% (Median [IQ]). Single cells were sorted on a FACS Aria III cell sorter (BD) following staining and suspension in an EDTA-containing sorting buffer. Quality of sorting was assessed using the staining index from the DIVA software version 8.0.

## Cytotoxicity assay

Effector cells were pre-incubated for 1 hr with or without 100 nM Concanamycin A (Sigma-Aldrich-Merck, Germany). RAW target cells were labelled with PKH-26 (Sigma-Aldrich-Merck, Germany) as previously described (*Sheehy et al., 2001*). Effector cells were added to 5000 RAW cells at appropriate effector/target ratios and incubated for 4 hr in the presence of 2 µg/ml mouse anti-human CD3 antibody (clone OKT3). RAW cells incubated with the anti-CD3 antibody but without effector cells were used as controls. Percentage of lysis was calculated as the percentage of caspase3-positive RAW cells after subtraction of the % of active caspase3 in the control wells (*He et al., 2005*).

## Quantitative PCR

qPCR was performed using the Taqman RNA Amplification kit or the LightCycler Multiplex RNA Virus Master and a LightCycler 480 instrument (Roche). Raw data were analyzed using the fit points method and fold change was calculated with the Delta-Delta Cp method using the housekeeping gene *EEF1A1* (*EF1*) as a reference. Primers and fluorescent probes were designed using Primer3 and purchased from Eurogentec. A Taqman Gene Expression assay was used for *RUNX3* (Hs00231709_m1, Thermo Fisher) and *PLZF* (*ZBTB16*, Hs00232313_m1, Thermo Fisher) analyses. Oligonucleotide sequences are presented as *Supplementary file 2c*.

## Bisulphite pyrosequencing

The *PRF1* promoter was divided into 11 amplicons covering 34 CpG sites as previously described (*Narasimhan et al., 2009*). Genomic DNA was bisulphite-converted using the Qiagen FAST Epitect bisulfite kit and sequenced using a Pyromark Q96 device after isolation of single strand biotinylated DNA from the PCR product using streptavidin and a pyromark Q96 vacuum prep station. Quality assessment and methylation level calculation were performed using the software Pyro Q-CpG and the CpG assay 1.0.9 (Biotage).

## Nucleic acid material

DNA and RNA were extracted using the Qiagen AllPrep DNA/RNA kit. Concentration and purity were assessed by spectrophotometry (nanodrop - Thermoscientific). Median [interquartile range] of A260/A280 ratios were 1.93 [1.77–2.10] and 1.82 (1.71–1.91) for RNA and DNA samples, respectively. For microarray analyses, RNA integrity number (RIN) was measured using the Eukaryote Total RNA Nano assay and a Bioanalyzer (Agilent). One sample out of 25 had a RIN below seven and was excluded from the analyses.

## Gene expression and methylation microarrays

Total RNA was amplified with the Illumina TotalPrep RNA Amplification Kit (Ambion) and hybridized with the HumanHT-12 v4 array containing 47,323 probes for 44,053 annotated genes, according to

the instructions of the Whole-Genome Gene Expression Direct Hybridization Assay (Illumina). Chips were scanned with the HiScan Reader (Illumina). For methylation analyses, genomic DNA was bisulphite-converted using an EZ DNA methylation Kit (ZYMO). DNA methylation level was measured using the Illumina Infinium HD Methylation Assay. Bisulphite converted DNA was hybridized with the Illumina HumanMethylation450 BeadChip 450K array (12 samples/chip), as described previously (*Dedeurwaerder et al., 2011*). Data from both arrays are available on GEO (https://www.ncbi.nlm.nih.gov/geo/) under the accession number GSE75406.

## Chromatin immunoprecipitation (ChIP) and ChIP-qPCR

MACS-purified CD4 T cells were stained and fixed with 1% formaldehyde. Glycine was added at a final concentration of 0.125 M to quench the crosslinking reaction. Cells were washed twice with ice-cold PBS and resuspended in complete medium for FACS sorting. Dry pellets of sorted cells were frozen at −80°C. Thawed pellets were lysed in 1% SDS-containing buffer and sonicated to obtain DNA fragments of 300–800 bp using a Bioruptor device (Diagenode). Chromatin of 200,000 cells was immunoprecipitated with an anti-histone antibody and protein G magnetic-activated beads. Chromatin was incubated overnight at 4°C with the following antibodies: 1 µg anti-H3K4me3 (Millipore 17–614 rabbit monoclonal), 0.5 µg anti-H3K27ac (abcam ab4729 rabbit polyclonal) or anti-H3 (diagenode C15310135 rabbit polyclonal). 1% of the IP reaction was collected before adding the antibody and the beads and was used as total chromatin input. Beads were washed five times: once with low-salt buffer, once with high-salt buffer, once with lithium chloride containing buffer and twice with TE buffer. After washing and reverse crosslinking (incubation with NaCl 200 mM for 4 hr at 65°C), chromatin was eluted with a buffer containing 1% SDS and 100 mM NaHCO$_3$ and treated with RNAse A and Proteinase K for 1 hr at 45°C. IP-DNA was purified using the MinElute PCR purification kit (Qiagen) and then analyzed by qPCR using the Probe Master 480 kit with primers encompassing different regulatory regions of the perforin locus (sequences are presented as *Supplementary file 2d*). The DeltaCp method was used to calculate the % of input for each IP. Results were normalized for the DeltaCp of H3-IP DNA.

## Singe-cell qPCR assay

Single cells were collected in 5 µL lysis buffer (CelluLyser micro lysis buffer from Tataa biocenter), immediately frozen on dry ice and stored at −80°C until used (*Svec et al., 2013*). Reverse transcription (RT) was performed using the CelluLyser Micro Lysis and cDNA Synthesis Kit following manufacturer's instructions (Tataa biocenter). RT step was validated using the Universal RNA Spike (TATAA Universal RNA Spike I from Tataa biocenter) in each well to ensure the absence of inhibitions. Wells most likely to contain single cells were selected on the basis of housekeeping gene expression and exclusion of outliers. cDNA was then pre-amplified for 20 cycles using the TATAA PreAmp GrandMaster Mix kit from the same company. Single-cell qPCR was performed on 43 cells per subset in 96.96 Dynamic Array IFC plates for Gene Expression (BMK-M-96.96) using the fluidigm technology (Biomark HD). Primer sequences are presented in *Supplementary file 2e*.

## In vitro activation and polarisation of naive CD4 T cells

Ex vivo isolated naive CD4 T cells were activated with 0.5 µg/ml phytohemagglutinin (PHA) in the presence of allogeneic CD4-depleted irradiated PBMC used as feeders at the ratio of 1/1. Cells were then cultured in the presence of IL-2 (R and D, 6 ng/ml), IL-15 (R and D, 5 ng/ml) and Th1, Th2 or Th17 polarising cytokines for 3 to 21 days. Culture medium was RPMI 1640 (Lonza) supplemented with 10% Fetal calf serum, 1% amino acids and glutamine, 1% Penicillin/Streptomycin. All incubation steps were performed at 37°C with 5% CO$_2$. IL-2 and IL-15 were added on day 2 and medium was replenished when required on the basis of cell proliferation. Polarizing cytokines and neutralizing antibodies were purchased from eBiosciences and used at final concentration of 10 ng/ml and 10 µg/ml, respectively. Th1 polarizing medium contained IFNγ, IL-12, anti-IL-4 (clone MP4-25D2), and anti-IL-17 (clone eBio64CAP17). Th2 polarizing medium contained IL-4, anti-IL 17, anti-IFNγ (clone NIB42), and anti-IL-12 (clone 20C2). Th17 polarizing medium contained IL-1β, IL-6, IL-23, TGFβ, anti-IL-4, anti-IFNγ, and anti-IL-12. On day 3, CD4 T cells were isolated by MACS-positive selection before nucleic acid extraction. On days 7, 14 and 21, cells were collected for downstream analyses.

## Transcription factor knockdown

Lentiviral particles were produced by transient transfection of packaging HEK 293 T cells. pMD2.G and psPAX2 were used as envelop and core packaging plasmids, respectively, together with the gene transfer plasmid (*Supplementary file 2f*). Before transduction, 50,000 freshly isolated naive CD4 T cells were stimulated during 46 hr in the presence of feeders and PHA (5 µg/ml) in Th1-cytokines containing X-Vivo15 medium (Lonza). For transduction, viral particles were added at a MOI of 10, together with IL-2 and IL-15 in 50 µl of fresh medium. Cells were amplified during 10 days before sorting of GFP$^+$ transduced cells for down-stream analysis.

## TCR CDR3 sequencing

Purified cDNA (AMPure XP Beads (Agencourt)) was obtained from total RNA and used in template-switch anchored RT-PCR experiment with specific alpha and beta chain primers. PCR products were then submitted to high-throughput sequencing as previously described (*Van Caeneghem et al., 2017*). Briefly, V2 300 kit with 200 bp at the 3' end (read 2) and 100 bp at the 5' end (read 1) were used on the Illumina MiSeq platform.

## Statistical analyses

Data were analyzed with the GraphPad Prism software unless otherwise specified. After one-way ANOVA analysis of variance, a Mann-Whitney test was performed for selected two-by-two comparisons and a Dunnet's test for multiple comparisons when appropriate. For grouped analysis, we used two-way ANOVA with multiple Tukey's tests. Differences were considered statistically significant at p-values<0.05.

### Illumina Expression HT12 Arrays

Raw data were quantile normalized using the normalization method from the lumi package (*Du et al., 2008*). Unsupervised clustering (Uc) analysis of gene expression datasets was performed using the pvclust package of the R software (R)(*Suzuki and Shimodaira, 2006*). The robustness of the Uc tree was tested by multiscale bootstrap resampling using Pearson's correlation as distance and Ward.D2 as clustering method, with 1000 iterations. An AU (approximately unbiased) p-value (percentage) was calculated and placed on the nodes of the cluster dendrogram. Principal component analysis (PCA) on the expression dataset was performed using the MultiExperiment Viewer (MeV) tool and the scatter plot function in R. The GeneSign module of the BubbleGUM software (*Spinelli et al., 2015*) was used with the Min/max method to identify lists of genes specifically expressed in cell subsets. A probe was considered specific of a given cell subset if its minimal expression value across the replicates of the cell subset of interest (test population) was higher than its maximal expression value across the replicates of the cell population used as reference (reference population). To obtain a limited number of genes, probes for which the ratio between the maximal and minimal expression values across all samples was below 1.2 were considered not regulated in any cell subsets and thus excluded from the analysis. Finally, probe lists were transformed into gene lists (or GeneSets). Heatmaps were generated using the heatmap.2 function of gplots package of R.

### Illumina HumanMethylation450 BeadChip arrays

Raw data were filtered using a detection p-value<0.05. Cross-reactive probes were filtered out while probes containing SNPs, which do not introduce an important confounder in intra-individual studies, were kept in the analysis as previously detailed.(*Dedeurwaerder et al., 2014*) β-values were computed using the following formula: β-value = M/[U + M] where M and U are the raw 'methylated' and 'unmethylated' signals, respectively. The β-values were corrected for type I and type II bias using the peak-based correction (*Dedeurwaerder et al., 2011*; *Dedeurwaerder et al., 2014*). The differential analyses were applied according to published recommendations (*Dedeurwaerder et al., 2014*): first the methylation values were converted to M-values using the following formula: M-value = log2 (β-value/(1–β-value)). The statistical significance of the differential methylation was then assessed using a paired t-test applied on these M-values. In parallel, a Delta-β was computed as the absolute difference between the median β-values. Cytosine showing a p-value<1e-4 together with an absolute delta-β>0.1 were reported as differentially methylated. Heatmaps were generated based on the scaled beta values of all the probes located in the promoter region of each

represented gene, using the heatmap.2 function of gplots package of R. Promoter regions included 5'prime, TSS1500, TSS200 and 1st exon.

## Gene Set Enrichment Analysis (GSEA)

GSEA was used to analyze the enrichment of GeneSets obtained by GeneSign on the pairwise comparisons of our gene expression microarray data (*Subramanian et al., 2005*). Catalog c3 from MsigDB was added to our GeneSets for robust statistical analysis. GSEA was used with 1000 GeneSet-based permutations and 'difference of classes' as a metric for ranking the genes since the expression values were in Log2 scale. The software quantifies enrichments by computing the Normalized Enrichment Score (NES) and the False Discovery Rate (FDR). FDR below 0.25 was considered significant (*Subramanian et al., 2005*). Genes showing no differential expression between $CD8_{CTX}$ and $CD4_{CTX}$ were used as negative control. GSEA was also used to quantify the correlation between gene expression and methylation. The identifiers of the genes included in the GeneSets were transformed to match the DNA methylation probe identifiers (Perl scripts included in *Supplementary file 3*). GSEA Pre-ranked analysis was then used to assess the enrichment of our expression-based GeneSets on the pairwise comparisons of the pre-ranked methylation delta β values. Genesets of catalog c3 from MsigDB were similarly converted to probe identifiers and added to our GeneSet file for robust statistical evaluation.

## Single-cell qPCR

Data mining was performed using the Fluidigm Real-time PCR analysis (V1.4.3), and data analysis was performed using the Genex6 MultiD software as previously described (*Ståhlberg et al., 2013*). Heat Map was created using the standard function in R.

## TCR repertoire

Raw sequencing reads from fastq files (both reads) were aligned to reference V, D and J genes specifically for 'TRA' or 'TRB' to build CDR3 sequences using the MiXCR software version 1.7. (*Bolotin et al., 2015*). CDR3 sequences were then assembled and clonotypes were exported and analyzed using the tcR package (*Nazarov et al., 2015*).

## Acknowledgements

We thank the blood donors and ImmuneHealth for the clinical sample collection. We are grateful to Véronique Olislagers, Sandra Lecomte, Margreet Brouwer, Muriel Nguyen, Séverine Thomas, Inès Vu Duc, Charlotte Givord and Iain Welsby for technical assistance, Ariane Huygens for precious expertise in flow cytometry and Charles Dehout, Nicolas Dauby, Muriel Moser, Alain Le Moine and Fabienne Willems for helpful discussions, Sophie Lucas (Institut de Duve – Université Catholique de Louvain) for providing single cell PCR reagents as well as Thierry Voet and Koen Theunis (Katholieke Universiteit Leuven) for single cell PCR analyses. YS is research assistant, SG senior research associate and AM research director at the Fonds de la Recherche Scientifique (FRS – FNRS), Belgium. A.H. was supported by the ASCO grant of the Fonds Erasme. This work was supported by the FRS. – FNRS, by the Belgian Federal Public Planning Service Science Policy and by the European Regional Development Fund (ERDF) and the Walloon Region (Wallonia-Biomed portfolio, 411132–957270).

## Additional information

### Funding

| Funder | Grant reference number | Author |
| --- | --- | --- |
| Fonds De La Recherche Scientifique - FNRS | PhD Student Fellowship | Yasmina Serroukh |
| Belgian Federal Public Planning Service Science Policy | Research Project Grant | Stanislas Goriely<br>Arnaud Marchant |

| European Regional Development Fund and Walloon Region | Research Project Grant (411132-957270) | Stanislas Goriely Arnaud Marchant |
|---|---|---|
| Fonds De La Recherche Scientifique - FNRS | Research Project Grant (PDR) | François Fuks Stanislas Goriely Arnaud Marchant |
| Fonds Erasme | PhD Student Fellowship | Alice Hoyois |

The funders had no role in study design, data collection and interpretation, or the decision to submit the work for publication.

### Author contributions

Yasmina Serroukh, Chunyan Gu-Trantien, Baharak Hooshiar Kashani, Conceptualization, Formal analysis, Investigation, Writing—original draft, Writing—review and editing; Matthieu Defrance, Thien-Phong Vu Manh, Jishnu Das, Martin Bizet, Emeline Pollet, Formal analysis, Writing—review and editing; Abdulkader Azouz, Aurélie Detavernier, Alice Hoyois, Tressy Tabbuso, Emilie Calonne, Investigation, Writing—review and editing; Klaas van Gisbergen, Marc Dalod, François Fuks, Conceptualization, Resources, Writing—review and editing; Stanislas Goriely, Arnaud Marchant, Conceptualization, Supervision, Funding acquisition, Writing—review and editing

### Author ORCIDs

Yasmina Serroukh  http://orcid.org/0000-0001-8751-5725
Thien-Phong Vu Manh  http://orcid.org/0000-0002-0294-342X
Jishnu Das  http://orcid.org/0000-0002-5747-064X
Stanislas Goriely  http://orcid.org/0000-0002-7005-6195
Arnaud Marchant  http://orcid.org/0000-0003-0578-0467

### Ethics

Human subjects: Volunteers were recruited by the research centre ImmuneHealth, CHU Tivoli, La Louvière. Clinical staff informed the volunteers about the objectives of the study and obtained their written consent to use the human material for research purposes. The study and the informed consent form were approved, following approval by the Ethics committee of the CHU Tivoli, La Louvière, Belgium (Reference B09620097253). The study followed the Good Clinical Practice (ICH/GCP) guidelines, the Belgian Law and the declaration of Helsinki ("World Medical Association Declaration of Helsinki; Ethical Principles for Medical Research Involving Human Subjects").

### Decision letter and Author response

Decision letter https://doi.org/10.7554/eLife.30496.035
Author response https://doi.org/10.7554/eLife.30496.036

## Additional files

### Supplementary files

• Supplementary file 1. Lists of genes included in the 12 GeneSets obtained from CD4 versus CD8 T cell comparison (*Figure 2*) and naive CD4 T cell versus CMTh1 cell versus CD4CTX T cell comparison (*Figure 4*). Genes expressed at higher level by the first listed subset as compared to the subset indicated between brackets were identified using the min/max method. Genes are ranked according to mean log2 fold change calculated using the Limma package in R.
DOI: https://doi.org/10.7554/eLife.30496.026

• Supplementary file 2. (a) Level of methylation at individual CpG sites in memory Th1 cell subsets in vivo. The level of methylation was measured by pyrosequencing in memory Th1 cell subsets of two to nine CMV-seropositive healthy adults. Data are median percentage ± interquartile range and were compared with the Mann-Withney non-parametric test. † $CM_{Th1}$ versus naive CD4 T cells. ‡ $EM28^{+}_{Th1}$ versus naive CD4 T cells. § $CD4_{CTX}$ versus naive CD4 T cells. ¶ $EM28^{+}_{Th1}$ versus $CM_{Th1}$. ‖ $CD4_{CTX}$ versus $EM28^{+}_{Th1}$. nd : not done, no statistical analysis was performed because of insufficient

number of subjects (n = 2 or 3) ; ns : non significant ;*=p<0.05 ; **=p<0.01 ; ***=p<0.001 ; ****=$p$<0.0001. (b) Monoclonal antibodies used in flow cytometry experiments. (c). QPCR Oligonucleotide sequences (d) ChIP-PCR oligonucleotide sequences. (e) Single-cell qPCR oligonucleotide sequences. (f) References for the gene transfer plasmids used in transcription factor knockdown experiments.

DOI: https://doi.org/10.7554/eLife.30496.027

• Supplementary file 3. Method to analyze the correlation between gene expression and DNA methylation. Gene Set Enrichment Analysis (GSEA) was used to analyze the enrichment of DNA methylation profiles in the GeneSets expressed by T cell subsets. Five Perl scripts were created to prepare the file required for GSEA.

DOI: https://doi.org/10.7554/eLife.30496.028

• Transparent reporting form

DOI: https://doi.org/10.7554/eLife.30496.029

### Major datasets

The following dataset was generated:

| Author(s) | Year | Dataset title | Dataset URL | Database, license, and accessibility information |
|---|---|---|---|---|
| Serroukh Y, Kashani BH, Gu-Trantien C, Defrance M, Vu Manh TP, Azouz A, Detavernier A, Das J, Bizet M, Pollet E, Tabbuso T, Calonne E, van Gisbergen K, Dalod M, Fuks F, Goriely S, Marchant A | 2017 | Genome-wide study of transcriptional and methylation profile of human cytotoxic CD4 T cells | https://www.ncbi.nlm.nih.gov/geo/query/acc.cgi?&acc=GSE75406 | Publicly available at the NCBI Gene Expression Omnibus (accession no: GSE75406) |

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
