## [Decision Letter]

Thank you for submitting your article "The transcription factors Runx3, T-bet and ThPOK co-regulate acquisition of cytotoxic function by human Th1 lymphocytes" for consideration by *eLife*. Your article has been reviewed by three peer reviewers, and the evaluation has been overseen by a Reviewing Editor and Tadatsugu Taniguchi as the Senior Editor. The following individual involved in review of your submission has agreed to reveal his identity: Remy Bosselut, whose comments have been incorporated with those of the other referees below.

The reviewers have discussed the reviews with one another and the Reviewing Editor has drafted this decision to help you prepare a revised submission.

Serroukh et al. investigate the transcriptional control of human CD4 cytotoxic cell differentiation. They demonstrate that these cells express transcription factors Runx3, Eomes and T-bet, and maintain high levels of the CD4-differentiating factor Thpok. The latter result contrasts with previous studies on mouse IELs by the Cheroutre and Mucida groups. Combining in vitro cytotoxic CD4 T cell differentiation and shRNA knock-down, the manuscript further shows that T-bet and Runx3, but not Eomes, are important for the acquisition of cytotoxic properties. Conversely, in agreement with mouse studies (including NI, 15:947), Thpok knock-down increases expression of T-bet, Runx3, Eomes and CD8α.

CD4 cytotoxic cells are increasingly focusing interest because of their frequency in human viral infections and their potential in anti-tumor responses. Thus, the present investigation of their transcriptional landscape is timely and should be of interest to a broad audience. The data are of high quality and well presented. While many results recapitulate observations in mouse models, others, including the persistent Thpok expression, would not have been predicted from such studies. Additionally, the extensive analysis of gene expression and DNA methylation in human effector T cells will be a useful resource.

Major comments requiring attention:

1) Various sets of data related to key findings could be improved by adding extra information or quantification. For example, flow cytometry profiles of transcription factors are usually not very clear. The authors use this technique a few times during their work. In particular, in Figure 2 ThPOK level in CD8 CTX mRNA does not match the protein level as well as EOMES in the CD8 N. Do the authors have an explanation for this? It is also vital to introduce some sort of quantification when analyzing the protein level of these transcription factors. It would be advisable to add the respective MFIs on the plots.

2) The non-essential effect of Eomes in CD4_CTX_ cells is an interesting finding. The authors should further build on this point through comparing the relative importance of T-bet and Eomes in CD4_CTX_ and CD8_CTX_ cells. The authors should also clarify whether ThPOK is indeed expressed by CD8_CTX_ cells and whether knocking-down ThPOK in CD8_CTX_ cells has any effect on their cytotoxic function.

3) Even though they differentiate in vitro from bulk populations, one key question is whether the ex vivo CD4 cytotoxic effectors are conventional MHC-II restricted cells, or include non-conventional T cells. Analyses of TCR Vβ expression are needed to address this question, and would also provide insight into the clonality of ex vivo cytotoxic cells. Staining for PLZF expression would also be informative. It is not clear whether the "CD4_CTX_" cells generated in vitro under Th1 culture conditions can represent CD4_CTX_ cells found in vivo. Have the authors compared the transcriptomes of these cells? The authors should try to knockdown Runx3, T-bet etc. in CD4_CTX_ cells isolated from PBMC to assess the effect on cytotoxic function.

4) RT-PCR or staining for Eomes is needed to resolve the contrast between Figure 5 (Eomes up-regulated in cytotoxic CD4s by microarray) and Figure 4 (mutually exclusive with perforin in single cell analyses).

5) Eomes knock-down is actually modest (Figure 7—figure supplement 1), and the conclusion that cytotoxic gene expression in CD4 cells is Eomes-independent should be toned down. Because shRNA experiments are notoriously difficult and prone to off-target effects, the study would be much strengthened if the authors verified shRNA impact on their target by immunoblot analyses of protein expression rather than RT-PCR or the rather inconclusive flow cytometry data shown in Figure 7—figure supplement 1.

6) The conclusion that stepwise changes in the chromatin environment promote expression of cytotoxic genes should be tempered. The supporting data is correlative only and the population-level changes in DNA methylation and histone modifications may simply reflect increased fractions of cytotoxic cells in each population.

7) In Figure 2, authors only compared CD4_CTX_ and CD8_CTX_ cells with naïve CD4 and CD8 T cells. Other memory CD4 and CD8 T cells, such as CMTh1, EMTh1 and CD27+ CD8 T cells, should be included for comparison to reach a solid conclusion on similarity between CD4_CTX_ and CD8_CTX_ cells, at least for Figure.2A, 2B and 2D. These cells should also be included as controls in the cell lysis assay in Figure 1.

8) Is CD28 downregulated in perforin+ Th1 cells generated in vitro compared to perforin- Th1 cells in the same culture? If so, the authors should separate CD28+ and CD28- Th1 cells to test their progressive development model. An alternative explanation for increased Granzyme+Perforin+ cells over time is that these cells were selectively expanded.

9) It is surprising (possibly important and novel) that Eomes does not play an important role in the acquisition of cytotoxicity by CD4_CTX_ cells. Can this shRNA affect the cytotoxicity of CD8_CTX_ cells? Are Runx3 and T-bet important for CD8_CTX_ cells? It is also curious that Eomes mRNA does not co-express with PRF1 in CD4_CTX_ cells. Co-staining of Eomes and PRF1 protein should be shown.

10) In the beginning of the study the authors define two major populations of CD4 T cells, naïve and cytotoxic. However, it seems the naïve subset is defined using CD45RO and the cytotoxic using perforin. There is no reference or explanation on why the authors use CD45RO. They should be consistent in using Perforin or CD45RO to distinguish between these two populations and ideally identify such populations in Figure 1, and not only in Supplementary Figure 1.

11) In Figure 3 it is not clear why the authors use EM CD8^+^ T cells versus total CD4. Furthermore, the figure legend only mentions the ChIP done on the CD4 subsets. This CD8 population is never defined and the total CD4 population is very heterogeneous, limited the conclusions that can be drawn by the authors. In this regard, the ChIP peaks should be shown for the different populations of CD4 T cells that are further shown in the plots with the normalized enrichment.

12) It would be suitable that a further quantification of the protein level of perforin would be depicted upon the knockdown of the different subsets in Figure 7. A simple set of representative plots is not enough to draw proper conclusions.

[Editors' note: further revisions were requested prior to acceptance, as described below.]

Thank you for resubmitting your work entitled "The transcription factors Runx3 and ThPOK cross-regulate acquisition of cytotoxic function by human Th1 lymphocytes" for further consideration at *eLife*. Your revised article has been favorably evaluated by Tadatsugu Taniguchi (Senior Editor), a Reviewing Editor, and three reviewers.

The manuscript has been improved but there are some remaining issues that need to be addressed before acceptance, as outlined below. As you can see, these focus on the issue of quantification of some of the key data added to the paper. We hope that these issues can be rapidly addressed without additional experimentation and, provided the quantification supports the claims made, the revised paper can be rapidly reviewed by the editor for a final decision.

*Reviewer #1:*

[Minor comments not shown.]

*Reviewer #2:*

The authors have addressed my previous concerns.

*Reviewer #3:*

The authors have extensively revised their original manuscript providing now a much more compelling and complete story. In this revised version, the authors have addressed all my previous concerns. There are however two main points that require further attention.

In Figure 2 the authors have added the much needed quantification of the expression of the different transcription factors in the populations of interest. This is represented by a FACS plot and the respective MFI quantification in a dot plot. Yet, the FACS plots are lacking the respective gates that give rise to the MFI quantification. It would vital that these gates are illustrated in Figure 2. Specially, since TBX21 and EOMES expression in CD8s appears to be across 3 populations for perforin expression, making it unclear which one was considered CD8 N and CD8_CTX_.

Additionally, in Figure 2 the authors claim that there was no apparent difference in the methylation of the ZBTB7B promoter between CD4_CTX_ and CD8_CTX_. The data shown is inconclusive and subjective to the reader's interpretation. If anything it appears that ZBTB7B promoter is more hypomethylated in CD4_CTX_ compared to CD8_CTX_ but without any sort of quantification all of these conclusions are very individual specific. The authors conclusions on this piece of data need to be tempered down.

[Minor comments not shown.]

---

## [Author Response]

Major comments requiring attention:1) Various sets of data related to key findings could be improved by adding extra information or quantification. For example, flow cytometry profiles of transcription factors are usually not very clear. The authors use this technique a few times during their work. In particular, in Figure 2 ThPOK level in CD8 CTX mRNA does not match the protein level as well as EOMES in the CD8 N. Do the authors have an explanation for this? It is also vital to introduce some sort of quantification when analyzing the protein level of these transcription factors. It would be advisable to add the respective MFIs on the plots.

We thank the reviewers for this helpful suggestion. We now provide protein expression quantification data in Figure 2Ee, Figure 3, Figure 6, Figure 7, Figure 7, Figure 6—figure supplement 2 and Figure 7—figure supplement 1. Overall, these data confirm the conclusions proposed in our initial manuscript. In particular, ThPOK and Eomes protein expression data confirmed mRNA expression data in CD8 T cell subsets (Figure 2). The only observed discrepancy relates to ThPOK expression in CD4 T cell subsets where mRNA data indicate a higher expression in CD4_CTX_ as compared to naive T cells whereas protein data indicate a high and similar expression in both subsets. This difference may be related to post-transcriptional regulation of ThPOK gene expression or to a higher sensitivity of the quantitative PCR as compared to flow cytometry. We consider that both results confirm our conclusion that the differentiation of human CD4_CTX_ T cells is not associated with, and does not require the down regulation of ThPOK expression.

2) The non-essential effect of Eomes in CD4_CTX_ cells is an interesting finding. The authors should further build on this point through comparing the relative importance of T-bet and Eomes in CD4_CTX_ and CD8_CTX_ cells. The authors should also clarify whether ThPOK is indeed expressed by CD8_CTX_ cells and whether knocking-down ThPOK in CD8_CTX_ cells has any effect on their cytotoxic function.

As suggested by the reviewers, we examined the expression and role of ThPOK and Eomes in the in vitro differentiation of CD8_CTX_ T cells. In three independent experiments, CD8_CTX_ T cells expressed high levels of Eomes and ThPOK. ThPOK knockdown up regulated the expression of granzyme B but did not consistently modify the expression of perforin. These results suggest that ThPOK may limit the cytotoxic activity of CD8_CTX_ T cells through the regulation of granzyme B expression and are presented in Supplementary Figure 10. The results are also in line with the notion that ThPOK promotes the anti-viral functions of CD8 T cells and the differentiation of cytotoxic innate T lymphocytes (1-3).

In the same experiments, Eomes knockdown did not impact the expression of perforin or granzyme B. This negative result could be related to the relatively moderate knockdown of the transcription factor (see Author response image 1) and/or to a non-essential role of Eomes in our experimental system. It is important to note that, to our knowledge, the existing literature does not provide a direct demonstration of a role for endogenous Eomes in the differentiation of CD8_CTX_ T cells. Ectopic expression of Eomes is sufficient to induce cytotoxicity in both CD4 and CD8 T cells (Cheroutre and Husain, 2013) and the inhibition of the common sequence of T-bet and Eomes (T-box cassette) decreases perforin expression (5). However, to our knowledge, data on the impact of Eomes knockdown in mice or humans are lacking. We consider that this important point requires more in-depth analyses and that it would not be appropriate to include these results in our manuscript.

**Author response image 1. respfig1:** Influence of Eomes on the differentiation of cytotoxic CD8 T cells. Naive CD8 T cells were submitted to polyclonal stimulation with PHA in the presence of irradiated allogeneic feeder cells, IL-2, IL-15 and Th1 polarizing cytokines for 7 to 14 days. Perforin, granzyme B (GZMB) and Eomes expression was measured by flow cytometry and by qPCR. a) Differentiated CD8 T cells expressed perforin and a fraction of them also expressed Eomes. Gates were placed using an unstained control and numbers indicate the% of cells in each quadrant. Comparable results were obtained in three independent experiments. b) Suppression of Eomes expression in differentiated CD8 T cells was performed using specific shRNA. Eomes knockdown, verified at mRNA and protein levels, did not significantly influence the expression of perforin or granzyme B as compared to non-silencing (N-S) shRNA. Data are mean +/- SD log2 Fold change of 5 replicates from 3 independent experiments (upper panels) and one histogram with median fluorescence intensity (MFI) of 2 independent experiments. *:p<0.05.

3) Even though they differentiate in vitro from bulk populations, one key question is whether the ex vivo CD4 cytotoxic effectors are conventional MHC-II restricted cells, or include non-conventional T cells. Analyses of TCR Vβ expression are needed to address this question, and would also provide insight into the clonality of ex vivo cytotoxic cells. Staining for PLZF expression would also be informative. It is not clear whether the "CD4_CTX_" cells generated in vitro under Th1 culture conditions can represent CD4_CTX_ cells found in vivo. Have the authors compared the transcriptomes of these cells? The authors should try to knockdown Runx3, T-bet etc. in CD4_CTX_ cells isolated from PBMC to assess the effect on cytotoxic function.

Following the suggestion of the reviewers, we analysed TCR repertoire and PLZF expression analyses on in vitro differentiated CD4 T cells. TCR α and β CDR3 sequencing revealed that CD4_CTX_ T cells included a diverse repertoire of clonotypes with similar proportions of large and intermediate expansions as compared to Th17 and Th2 cells. Furthermore, CD4_CTX_ T cells expressed low levels of PLZF. These results indicate that in vitro differentiated CD4_CTX_ T cells are conventional and not innate-type effector T lymphocytes and are presented as Figure 6—figure supplement 1.

We consider that our analysis of the expression of cytotoxic molecules, cytokines, chemokine receptor and transcription factors by in vitro differentiated CD4_CTX_ T cells and of their perforin-dependent cytotoxic activity shows that these cells express key features of CD4_CTX_ differentiated in vivo. We did not perform transcriptome analysis of the in vitro differentiated cells as the strategy we followed was to concentrate our analyses on candidate transcription factors identified through the study of cells differentiated in vivo.

in vivo differentiated human CD4_CTX_ are terminally differentiated, have short telomere length (Brown et al., 2012) and are difficult to maintain in culture in vitro. Our attempts to grow sorted CD4_CTX_ T cells have not been successful. Therefore, we could not follow the suggestion of the reviewers to perform knockdown experiments with this cell population. The in vitromodel of CD4_CTX_ T cell differentiation was developed to overcome this constraint.

4) RT-PCR or staining for Eomes is needed to resolve the contrast between Figure 5 (Eomes up-regulated in cytotoxic CD4s by microarray) and Figure 4 (mutually exclusive with perforin in single cell analyses).

We now provide staining and protein quantification data of Eomes and perforin expression by CD4 T cells (Figure 2). These data confirm that Eomes is expressed at higher levels in CD4_CTX_ as compared to naive cells but they also indicate that the magnitude of this up regulation is lower than those of Runx3 and T-bet and does not reach the levels observed in CD8_CTX_. RT-PCR data were included in our initial manuscript (Figure 3) and indicated that, in contrast to perforin, the expression of Eomes mRNA was not higher in CD4_CTX_ as compared to CD28^+^ EM_Th1_ cells. Data presented in Figure 4 did not indicate a mutually exclusive expression of Eomes and perforin in CD4_CTX_ T cells but rather that only a fraction of CD4_CTX_ T cells expressed Eomes mRNA whereas all expressed perforin mRNA. We realize that the Results section and the figure legend were not sufficiently clear and that this generated some confusion. They have now been amended to indicate more clearly that data are proportions of perforin^+^ cells co-expressing individual transcription factors. We performed RT-PCR verification of Eomes expression to further validate our observation. A somewhat higher proportion of Eomes^+^ cells was detected in CD4_CTX_ T cells as compared to our initial experiment. This did not change the observation that a lower proportion of perforin^+^ CD4_CTX_ T cells co-expressed Eomes as compared to the other co-expressed transcription factors. Figure 4 has been amended to include these results. The colour code of the figure has also been amended, avoiding the use of white colour, to make it clearer.

5) Eomes knock-down is actually modest (Figure 7—figure supplement 1), and the conclusion that cytotoxic gene expression in CD4 cells is Eomes-independent should be toned down. Because shRNA experiments are notoriously difficult and prone to off-target effects, the study would be much strengthened if the authors verified shRNA impact on their target by immunoblot analyses of protein expression rather than RT-PCR or the rather inconclusive flow cytometry data shown in Figure 7—figure supplement 1.

We now provide quantification of transcription factors knockdown by flow cytometry and we increased the number of experiments to perform statistical analyses (Figure 7 and Figure 7—figure supplement 1). These results confirmed the knockdown of Eomes by specific shRNA. We acknowledge in our manuscript that the relatively low magnitude of the knockdown may have contributed to the absence of impact on perforin expression. We discuss these results as well as the Eomes/perforin co-expression data obtained in vivo and in vitro and we suggest that, together, these observations do not support an essential role of Eomes in the acquisition of cytotoxic function by human CD4 T cells.

6) The conclusion that stepwise changes in the chromatin environment promote expression of cytotoxic genes should be tempered. The supporting data is correlative only and the population-level changes in DNA methylation and histone modifications may simply reflect increased fractions of cytotoxic cells in each population.

We agree with the comment of the reviewers and we have amended our manuscript accordingly.

7) In Figure 2, authors only compared CD4_CTX_ and CD8_CTX_ cells with naïve CD4 and CD8 T cells. Other memory CD4 and CD8 T cells, such as CMTh1, EMTh1 and CD27+ CD8 T cells, should be included for comparison to reach a solid conclusion on similarity between CD4_CTX_ and CD8_CTX_ cells, at least for Figure 2. These cells should also be included as controls in the cell lysis assay in Figure 1.

Data on the expression of transcription factor mRNA by CM_TH1_ and EM_TH1_ cells were already included in our initial manuscript (Figure 3). Following the suggestion of the reviewer, we now also provide transcriptome data for CM_Th1_ cells in Supplementary Figure 4. These data indicate that the transcriptional program of CD4_CTX_ is closer than the one of CM_Th1_ cells to the transcriptional program of CD8_CTX_ T cells. We did not conduct transcriptome analysis of EM CD28^+^_Th1_ cells because the perforin expression profile of this subset was more variable between donors than that of CM_Th1_ cells. Also, we did not conduct cytotoxicity assays with CM_Th1_ and EM CD28^+^_Th1_ cells because these subsets did not show significantly different expression of perforin protein as compared to naive cells (Figure 3).

8) Is CD28 downregulated in perforin+ Th1 cells generated in vitro compared to perforin- Th1 cells in the same culture? If so, the authors should separate CD28+ and CD28- Th1 cells to test their progressive development model. An alternative explanation for increased Granzyme+Perforin+ cells over time is that these cells were selectively expanded.

The expression of CD28 was not downregulated by in vitro differentiated CD4_CTX_ T cells after 2 weeks (see Author response image 2). This observation is in line with results obtained in in vitro model of effector CD8 T cell differentiation and indicating that only incomplete down regulation of CD27 was detected after 6 weeks of cell activation (Wilkinson et al., 2012).

**Author response image 2. respfig2:** Stable CD28 expression upon 14 days in vitro stimulation. Naive CD4 T cells were subjected to polyclonal stimulation as described in the manuscript (Figure 6). Results are flow cytometry plots of CD28 and CCR7 expression 3, 7 and 14 days after stimulation. Numbers indicate the% of cells in each quadrant.

9) It is surprising (possibly important and novel) that Eomes does not play an important role in the acquisition of cytotoxicity by CD4_CTX_ cells. Can this shRNA affect the cytotoxicity of CD8_CTX_ cells? Are Runx3 and T-bet important for CD8_CTX_ cells? It is also curious that Eomes mRNA does not co-express with PRF1 in CD4_CTX_ cells. Co-staining of Eomes and PRF1 protein should be shown.

Answers to this comment are provided under comments 2 and 4. Our attempts to knockdown the expression of Runx3 and T-bet in in vitro differentiated CD8_CTX_ T cells with specific shRNA have not been successful. Also, we consider that a detailed analysis of the transcription factors involved in the in vitro differentiation of human CD8_CTX_ T cells goes beyond the scope of our manuscript.

10) In the beginning of the study the authors define two major populations of CD4 T cells, naïve and cytotoxic. However, it seems the naïve subset is defined using CD45RO and the cytotoxic using perforin. There is no reference or explanation on why the authors use CD45RO. They should be consistent in using Perforin or CD45RO to distinguish between these two populations and ideally identify such populations in Figure 1, and not only in Supplementary Figure 1.

We thank the reviewers for this helpful suggestion. We now explain more carefully the gating strategy of naive and cytotoxic CD4 and CD8 T cells and we illustrate this strategy in Figure 1.

11) In Figure 3 it is not clear why the authors use EM CD8^+^ T cells versus total CD4. Furthermore, the figure legend only mentions the ChIP done on the CD4 subsets. This CD8 population is never defined and the total CD4 population is very heterogeneous, limited the conclusions that can be drawn by the authors. In this regard, the ChIP peaks should be shown for the different populations of CD4 T cells that are further shown in the plots with the normalized enrichment.

The comment of the reviewers indicate that Figure 3 was not clear. The EM CD8 and total CD4 T cell data had been obtained from a public database and were meant to illustrate the regulatory regions in the perforin promoter. To avoid confusion, we now provide a simpler version of the figure that does not include data that were not generated through our study (Figure 3).

12) It would be suitable that a further quantification of the protein level of perforin would be depicted upon the knockdown of the different subsets in Figure 7. A simple set of representative plots is not enough to draw proper conclusions.

As suggested by the reviewers, we now provide protein quantification data of perforin knockdown. We have also increased the number of experiments to perform statistical analyses (Figure 7).

1) Setoguchi R, Taniuchi I, Bevan MJ. ThPOK derepression is required for robust CD8 T cell responses to viral infection. J Immunol. 2009 Oct 1;183(Wilkinson et al., 2012):4467–74.

2) Park K, He X, Lee H-O, Hua X, Li Y, Wiest D, et al. TCR-mediated ThPOK induction promotes development of mature (CD24-) gammadelta thymocytes. The EMBO Journal. EMBO Press; 2010 Jul 21;29(Fu et al., 2013):2329–41.

3) Wang L, Carr T, Xiong Y, Wildt KF, Zhu J, Feigenbaum L, et al. The sequential activity of Gata3 and Thpok is required for the differentiation of CD1d-restricted CD4+ NKT cells. European Journal of immunology. WILEY‐VCH Verlag; 2010 Sep;40(9):2385–90.

4) Eshima K, Chiba S, Suzuki H, Kokubo K, Kobayashi H, Iizuka M, et al. Ectopic expression of a T-box transcription factor, eomesodermin, renders CD4(+) Th cells cytotoxic by activating both perforin- and FasL-pathways. Immunol Lett. 2012 May 30;144(1-2):7–15.

5) Pearce EL, Mullen AC, Martins GA, Krawczyk CM, Hutchins AS, Zediak VP, et al. Control of effector CD8+ T cell function by the transcription factor Eomesodermin. Science. 2003 Nov 7;302(5647):1041–3.

6) van de Berg PJEJ, Griffiths SJ, Yong S-L, Macaulay R, Bemelman FJ, Jackson S, et al. Cytomegalovirus infection reduces telomere length of the circulating T cell pool. J Immunol. 2010 Apr 1;184(Wilkinson et al., 2012):3417–23.

7) Papagno L, Spina CA, Marchant A, Salio M, Rufer N, Little S, et al. Immune activation and CD8+ T-cell differentiation towards senescence in HIV-1 infection. PLoS Biol. 2004 Feb;2(Appay et al., 2002a):E20.

[Editors' note: further revisions were requested prior to acceptance, as described below.]

Reviewer #3:

The authors have extensively revised their original manuscript providing now a much more compelling and complete story. In this revised version, the authors have addressed all my previous concerns. There are however two main points that require further attention.In Figure 2 the authors have added the much needed quantification of the expression of the different transcription factors in the populations of interest. This is represented by a FACS plot and the respective MFI quantification in a dot plot. Yet, the FACS plots are lacking the respective gates that give rise to the MFI quantification. It would vital that these gates are illustrated in Figure 2. Specially, since TBX21 and EOMES expression in CD8s appears to be across 3 populations for perforin expression, making it unclear which one was considered CD8 N and CD8_CTX_.

We thank the reviewer for this comment and suggestion. Figure 2 has been amended to show the gated populations using the same color code as the one used for the other panels of the figure. The strategy used to gate the naive and cytotoxic subsets was the same as the one illustrated in Figure 1. This is now clarified in the legend of Figure 2.

Additionally, in Figure 2 the authors claim that there was no apparent difference in the methylation of the ZBTB7B promoter between CD4_CTX_ and CD8_CTX_. The data shown is inconclusive and subjective to the reader's interpretation. If anything it appears that ZBTB7B promoter is more hypomethylated in CD4_CTX_ compared to CD8_CTX_ but without any sort of quantification all of these conclusions are very individual specific. The authors conclusions on this piece of data need to be tempered down.

We agree with the reviewer that the sentence describing these results could be misinterpreted. As suggested, we revised the sentence and tempered down the conclusion.